# Beyond Gradient and Priors in Privacy Attacks: Leveraging Pooler Layer Inputs of Language Models in Federated Learning

## Abstract

Federated learning (FL) emphasizes decentralized training by storing data locally and transmitting only model updates, underlining user privacy. However, a line of work on privacy attacks undermines user privacy by extracting sensitive data from large language models during FL. Yet, these attack techniques face distinct hurdles: some work chiefly with limited batch sizes (e.g., batch size of 1), and others can be easily detectable. This paper introduces an innovative approach that is challenging to detect, significantly enhancing the recovery rate of text in various batch-size settings. Building on fundamental gradient matching and domain prior knowledge, we enhance the recovery by tapping into the input of the Pooler layer of language models, offering additional feature-level guidance that effectively assists optimization-based attacks. We benchmark our method using text classification tasks on datasets such as CoLA, SST, and Rotten Tomatoes. Across different batch sizes and models, our approach consistently outperforms previous state-of-the-art results.

## 1 Introduction

Language models trained under the Federated Learning paradigm play a pivotal role in diverse applications such as next-word predictions on mobile devices and electronic health record analysis in hospitals (Ramaswamy et al., 2019; Li et al., 2020). This training paradigm prioritizes user privacy by restricting raw data access to local devices and centralizing only the model's updates, such as gradients and parameters (McMahan et al., 2017). While the FL framework is created to protect user privacy, vulnerabilities still persist. In the realm of Computer Vision (CV), there has been significant exploration, especially regarding image reconstruction attacks (Geiping et al., 2020; Yin et al., 2021; Jeon et al., 2021). In contrast, the Natural Language Processing (NLP) domain remains largely uncharted (Balunovic et al., 2022; Gupta et al., 2022).

Recent studies have investigated vulnerabilities of training data in Federated Learning when applied to language models (Zhu et al., 2019; Deng et al., 2021). These researches generally fall into two categories: **Malicious Attack** and **Eavesdropping Attack**. Malicious attacks often originate from malicious servers, which distribute harmful parameter updates or alter model architectures to covertly acquire user data (Fowl et al., 2021; 2022; Boenisch et al., 2023). These actions deviate from the standard protocol of FL and are usually obvious and can be easily detected by examining predefined architectures or using real-time local feature monitoring (Fowl et al., 2022). On the other hand, eavesdropping attacks do not require model modification, making them harder to detect. Adhering to the *honest-but-curious* principle, attackers leverage gradient information and prior knowledge to extract sensitive data (Zhu et al., 2019; Deng et al., 2021; Balunovic et al., 2022; Gupta et al., 2022). However, their efficacy is contingent on conditions like minimal batch sizes, with performance degradation as batch size grows (Balunovic et al., 2022). Contrasting with these findings, our research incorporates only minimal modifications to the model without violating the protocol of FL. This means it adheres to the ***honest-but-curious*** principle, making it challenging to detect and remarkably increase the effectiveness of the attack.

**Improved text privacy attack by leveraging unique feature information** Upon examining the current vulnerabilities in FL, the community has identified an issue with the gradient-based attack:

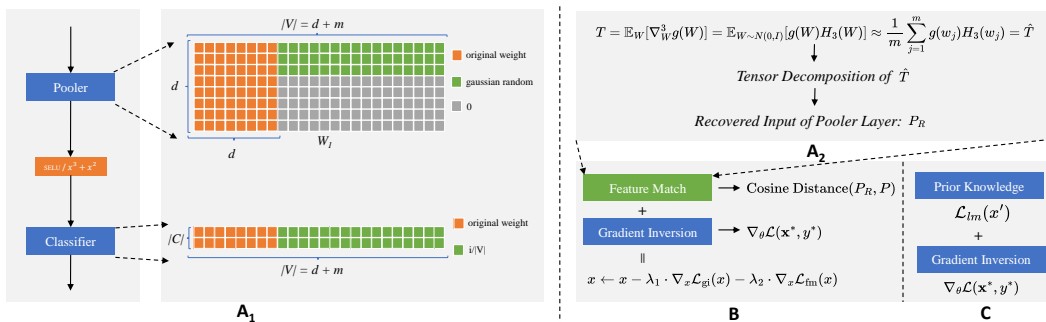

Figure 1: Architecture overview of our proposed attack mechanism on language models. $\mathbf{A}_1$: Subtle modification of architecture and strategic weight initialization. $\mathbf{A}_2$: Two-layer-neural-network-based reconstruction. **B**: Continuous optimization with gradient inversion and feature match. C: Discrete optimization with gradient matching loss and perplexity from pre-trained language models.

the gradient will be averaged in the context of large batch sizes and long sentences, thereby diluting the embedded information and reducing the attack's effectiveness. To address this challenge, we propose an innovative solution by recovering the intermediate feature to provide enhanced supervisory information. Specifically, we focus on Transformer-based language models equipped with a unique Pooler layer. This layer handles the final hidden state of the [CLS] token, capturing a comprehensive representation of the input text. Subsequently, we employ a two-layer-neural-network-based reconstruction technique to retrieve the inputs destined for this layer meticulously. In this way, our method introduces a fresh continuous supervisory signal besides gradients by leveraging the recovered intermediate features as a reference point. When combined with gradient inversion and prior knowledge, our approach consistently outperforms previous ones on a range of benchmark datasets and varied scenarios (where batch size $> 1$), underlining its resilience and versatility.

**Main Contributions**    Our main contributions are described as follows:

1. **Technical Contribution in Attack Method:** We are the first to suggest utilizing intermediate features as continuous supervised signals for text privacy attacks.

2. **Advancement in Intermediate Features Recovery:** We pioneered refining a two-layer-neural-network-based reconstruction method in practical deep language models, successfully recovering intermediate features.

3. **Superiority in Diverse Settings:** Our method consistently outperforms others across various benchmarks and settings by leveraging gradients, priors knowledge, and intermediate features, highlighting its robustness and adaptability.

4. **Semantic Evaluation Metric:** Recognizing the shortcomings of existing evaluation metrics, we introduce the use of semantic embedding distances as an alternative.

## 2    RELATED WORK

### 2.1    FEDERATED LEARNING

Introduced by McMahan et al. (2017), federated learning solves data privacy concerns by promoting decentralized model training. In this approach, models are refined using local updates from individual clients, which are then merged at a central server (Konečný et al., 2015; 2016; 2017). This field has attracted significant attention due to its potential business applications, underlining its relevance and promise in academia and industry (Ramaswamy et al., 2019; Li et al., 2020).

### 2.2    DATA PRIVACY ATTACK

While federated learning features with data privacy, recent studies show that model updates (gradients and parameters) can be intentionally leveraged to uncover sensitive data (Phong et al., 2017;

Zhao et al., 2020; Zhu & Blaschko, 2020; Zhu et al., 2019). This susceptibility is especially pronounced in the field of CV. In fact, some researchers have been able to recreate images almost perfectly by using gradients along with prior knowledge (Huang et al., 2021; Geiping et al., 2020; Yin et al., 2021; Jeon et al., 2021).

Textual data poses unique challenges in the context of private data attacks, especially given the prevalence of Transformer architectures. In Transformer, gradients average across sequences and tokens, which inherently masks specific token details. Furthermore, the inputs, expressed as discrete token IDs, starkly contrast the continuous features found in image data. Nonetheless, numerous studies have highlighted the risks associated with textual information. Current research on this topic can be broadly categorized into two groups.

**Malicious Attacks:** In this category, the central server has malicious intent. It may distribute networks with embedded backdoors or parameters that facilitate easy reconstruction of training data (Fowl et al., 2021; 2022; Boenisch et al., 2023). However, one can employ prefixed, recognized architectures to counter the former attack and guard against potential backdoor threats. For the latter attack, consistently monitoring statistics of features across different layers can help detect malicious parameter (Balunovic et al., 2022).

**Eavesdropping Attacks:** This approach assumes a trustworthy central server. Even with its integrity, the shared parameters and gradients could still be leveraged to extract private data (Zhu et al., 2019). For example, methods introduced by Zhu et al. (2019) and Deng et al. (2021) employ optimization-based strategies using finely-tuned objective functions for data retrieval. Balunovic et al. (2022) leverages prior knowledge from extensive language models for data recovery. However, due to the self-imported limitation (Server is benign without doing any change to the model), these methods tend to be less effective with larger batch sizes. Notably, the method introduced by (Gupta et al., 2022) remains effective even with considerable batch sizes. Nevertheless, this vulnerability can be easily defended by suspending updates to the language model's embedding matrix.

**Threat Model:** To further expose the risks to private textual data, we introduce a practical attack method that adheres to two principles: 1) We freeze the gradients of both token and positional embeddings. 2) We implement minimal changes to the model but do not deviate from the fundamental objectives of FL, such as effective gradient aggregation and model loss minimization. In our evaluation experiments, this method enhances the success rate of attacks across various batch sizes and datasets in the context of ***honest-but-curious*** setting.

## 3 PRELIMINARIES

### 3.1 GRADIENT INVERSION

Gradient inversion is a significant technique that could potentially breach privacy in federated learning (Zhu et al., 2019; Zhao et al., 2020). Although federated learning is designed to provide a decentralized training mechanism ensuring local data privacy, gradient inversion shows that this privacy may not be infallible.

**Problem definition:** Consider the supervised learning framework wherein a neural network $f(\cdot; \boldsymbol{\Theta}) : \boldsymbol{x} \in \mathbb{R}^d \to f(\boldsymbol{x}; \boldsymbol{\Theta}) \in \mathbb{R}$ is trained using the objective:

$$\min_{\boldsymbol{\Theta}} \sum_{(\boldsymbol{x},y)\in\mathcal{D}} \ell(f(\boldsymbol{x}; \boldsymbol{\Theta}), y) \tag{1}$$

where $\ell$ is the loss function and $\mathcal{D}$ denotes the dataset of input-output pairs. In the federated paradigm, during the communication between the central server and clients, each node reports an average gradient of its local batch $S$ (McMahan et al., 2017; Konečný et al., 2015). This is mathematically formulated as:

$$G := \frac{1}{B}\nabla_{\boldsymbol{\Theta}} \sum_{i=1}^{B} \ell\left(f\left(\boldsymbol{x}_i, \boldsymbol{\Theta}\right), y_i\right) \tag{2}$$

where $B$ is the batch size of $S$. Given the above definition, the challenge posed by gradient inversion becomes apparent: With access to a once-queried gradient and a known model as well as its loss function, is it possible to reconstruct the input training data?

**Objective of Gradient Inversion:** Based on the above problem, the objective of gradient inversion can be represented as:

$$\min_{\{\hat{\boldsymbol{x}}_i, \hat{y}_i\}_{i=1}^B} d\left(\frac{1}{B}\sum_{i=1}^B \nabla_{\boldsymbol{\Theta}}\ell\left(f\left(\hat{\boldsymbol{x}}_i; \boldsymbol{\Theta}\right), \hat{y}_i\right), G\right) \tag{3}$$

Here, $d(\cdot, \cdot)$ quantifies the difference between the provided and deduced gradients, and $(\hat{x}_I, \hat{y}_i)$ refers to the estimated input and its label. Prominent works have leveraged this objective to attempt the retrieval of private data (Zhu et al., 2019; Zhao et al., 2020; Geiping et al., 2020).

## 3.2 Employing Prior Knowledge for Text Recovery

Relying solely on gradient inversion to recover textual information often proves challenging, especially when handling larger batch sizes. Researchers often seek to acquire the prior knowledge encapsulated in pre-trained language models like GPT-2 to address this. These models are adept at predicting the probability of the next token based on the preceding sequence. This property aids in evaluating the quality of text searched through gradient inversion. Specifically, it introduces the perplexity as an evaluation metric to guide optimization by pinpointing optimal starting points or intermediate points of the attack (Balunovic et al., 2022; Gupta et al., 2022).

## 3.3 Two-layer-neural-network-based Reconstruction

Wang et al. (2023) demonstrates that it might be possible to reconstruct training data solely from gradient data using a theoretical approach within a two-layer neural network. Consider a two-layer neural network: $f(x; \Theta) = \sum_{j=1}^m a_j \sigma(w_j \cdot x)$, with parameters defined as $\Theta = (a_1, ..., a_m, w_1, ..., w_m)$. Here, $m$ represents the hidden dimension. The objective function is represented as: $L(\Theta) = \sum_{i=1}^B (y_i - f(x_i; \Theta))^2$. A notable finding is that the gradient for $a_j$ is solely influenced by $w_j$, making it independent from other parameters. This gradient is represented as:

$$g_j := \nabla_{a_j} L(\Theta) = \sum_{i=1}^B r_i \sigma\left(w_j^\mathsf{T} x_i\right) \tag{4}$$

where the residual $r_i$ is given by $r_i = f(x_i; \Theta) - y_i$. For wide neural networks with random initialization from a standard normal distribution, the residuals $r_i$ concentrate to a constant, $r_i^*$. By set $g_{(w)} := \sum_{i=1}^B r_i^* \sigma(w^\mathsf{T} x_i)$, $g_j$ can be expressed as $g_j = g(w_j) + \epsilon$, where $\epsilon$ represents noise. Then the third derivative of $g(w)$ is represented as:

$$\nabla^3 g(w) = \sum_{i=1}^B r_i^* \sigma^{(3)}(w^\mathsf{T} x_i) x_i^{\otimes 3} \tag{5}$$

Here, $x_i^{\otimes 3}$ signifies the tensor product of vector $x_i$ with itself three times. The researchers postulated that if they can accurately estimate $\nabla^3 g(w)$, it is possible to determine $\{x_i\}_{i=1}^B$ by using tensor decomposition techniques, especially when these features are independent. They used Stein's Lemma, expressed as: $\mathbb{E}[g(X)H_p(X)] = \mathbb{E}[g^{(p)}(X)]$ to approximate $\nabla^3 g(w)$ as:

$$T = \mathbb{E}_W[\nabla_W^3 g(W)] = \mathbb{E}_{W \sim N(0,I)}[g(W)H_3(W)] \approx \frac{1}{m}\sum_{j=1}^m g(w_j)H_3(w_j) = \hat{T} \tag{6}$$

Where $H_3(w_j)$ is the p-th Hermite function of $w_j$. By leveraging this approach, they successfully reconstructed each unique $x_i$. Their approach is predominantly theoretical and is mostly restricted to two-layer fully connected networks. Specifically, when applied to deeper networks, their method uses identity modules and other transparently detectable weight manipulations, which limits its practical use. In this work, instead of attempting to recover the input of a deep neural network directly, we aim to retrieve the intermediate features that serve as the subsequent optimization-based supervisory signals. Because we concentrate solely on a specific segment of the deep neural network, it becomes simpler to meet certain constraints. Further details will be provided in Section 4.2.

## 4 METHODOLOGY

This section first recalls the constraints of gradient inversion and suggests incorporating intermediate features as supervisory signals. We then delve into a feature reconstruction technique that employs a two-layer neural network, focusing the input to the Pooler layer in language models. Finally, we illustrate how to provide optimization signals with this recovered intermediate feature.

### 4.1 RECALL GRADIENT INVERSION

Gradient inversion seeks to reconstruct the original training data by harnessing the gradients of a known deep-learning model. A closer look at this method reveals several challenges. Central to these is the nonconvexity of the issue, marked by the presence of numerous local minima that complicate the pursuit of the global optimum. Additionally, the problem is over-determined because it has more equations to resolve than unknown parameters. While these equations remain consistent, they complicate the optimization process. This complexity persists even when reduced to a single-sample scenario. As a result, gradient inversion remains an NP-complete problem, implying that procuring an exact solution within a feasible time frame is difficult (Wang et al., 2023).

From a broader perspective, a distinct challenge arises. Language models typically handle batches of sentences, each containing multiple tokens. In the gradient inversion technique, the gradients of the entire batch are averaged. However, this doesn't only average the gradients for whole sentences but also for individual tokens within them. This process obscures the unique gradient patterns of each token, making their retrieval more complex. While the averaged gradient provides a general picture of the data, it conceals the finer details vital for precise token reconstruction. Given this intricacy, a pivotal question emerges: **Is it feasible to create an alternative method using a non-conventional, non-averaged supervisory signal, like feature information, to enhance data reconstruction?**

### 4.2 RECONSTRUCT INPUT OF POOLER LAYER

The previous study shows the possibility of recovering training data with only gradient information from a wide two-layer neural network (Wang et al., 2023). However, this research has many constraints as described in Section 3.3, and it does not restore the actual features but rather their direction within the feature space. Given these limitations, we've shifted from directly recovering the input of deep neural networks to exploring the potential of initially recovering their intermediate features. We are particularly drawn to the widespread use of the Transformer architecture in language models, with many (BERT, etc.) having a Pooler and Classifier at their head. As such, our approach aims to reconstruct the features directed to the Pooler layer, believing recovered intermediate features can offer a unique supervisory signal distinct from gradients and prior knowledge. While the aforementioned study provides foundational insights, our challenge was to adapt and refine this technique to match the unique demands of deep language models in practice.

**Subtle Modification of Architecture:** The initial configuration of language models often sets the hidden dimension of the Pooler layer to match the input dimension (For BERT$_{\text{BASE}}$, it's 768). This setting is insufficient to promise the accuracy of tensor decomposition when applying the two-layer-neural-network-based reconstruction method. To address this limitation, we expand the dimension of the Pooler layer to match the vocabulary size of the language models (For BERT$_{\text{BASE}}$, this was adjusted from 768 to 30,522). The rationale behind this change is grounded in enhancing the model's expressiveness while ensuring our modifications are not easily detectable.

Moreover, our empirical observations indicated that the original activation function struggles to work harmoniously with the recovery method, leading to inaccurate information retrieval. This challenge arises due to its $i_{th}$ derivatives resulting in zero expectations, expressed as $\mathbb{E}_{Z \sim N(0,1)}[\sigma^{(i_{th})}(Z)] = 0$, and leads to an inaccurate estimation of $\hat{T}$ as described in Equation 18. To counter these challenges, we replace the Tanh function after the Pooler layer with two alternative functions: SELU or $\sigma(x) = x^3 + x^2$. Neither of these functions is strictly odd or even, which counter issues from derivatives. It's worth noting that SELU, a commonly used activation function in deep learning, is less likely to draw attention. On the other hand, our empirical tests of the cube+square function indicate that while it compromises concealability, it offers enhanced attack performance in specific scenarios.

**Strategic Weight Initialization:** We introduce key notations first: $X$ is the input to the Pooler layer with a shape of $(B, d)$, where $B$ is the batch size and $d$ is the feature dimension. The weights $W_1$ and $W_2$ correspond to the Pooler and Classifier layers, respectively, with shapes $(|V|, d)$ and $(N, |V|)$. Here, $|V|$ is the vocabulary size and $N$ is the number of classification classes.

As mentioned in Section 3.3, $m$ signifies the hidden dimension in a two-layer neural network. Ideally, $|V|$ should be equivalent to $m$ in our setting. However, during our computation of $\hat{T}$ as outlined in Equation 18, we noticed an anomaly in $g_j$. Due to the random initialization of $W_1$, a substantial portion of $g_j$ approached a value close to 0. This side effect impacts the subsequent decomposition procedure. To address this issue, rather than setting $|V| = m$, we determined $m = |V| - d$. This approach ensures the remaining dimensions are randomly initialized and adequate to promise the accuracy of tensor decomposition. Simultaneously, the original weights are retained in the new weight matrix, allowing us to obtain optimal gradients for $W_1$ and $W_2$. For the classifier layer, we utilize a strategy similar to that of the Pooler layer, adjusting the remaining dimensions to a constant ($i/m$, where $i$ represents the class index for the classification task).

**Flexibility of the Recovered Dimension:** Wang et al. (2023) suggests significantly expanding the hidden dimension $m$ in comparison to the input dimension $d$ to reduce the tensor decomposition error. In our setting, we let $m = |V| - d$. Given that $|V|$ represents the vocabulary size, it sounds straightforward to utilize this value as the dimension of the Pooler layer. Any other configuration for $m$ appears less intuitive. Thus, it's reasonable for our choice, and there is no reason to adjust the hidden dimension any further. On the other hand, for a fixed $d$ (768 for BERT$_{\text{BASE}}$), determining the optimal value for $m$ can be challenging without adjustments. Recognizing these constraints, we kept $m$ constant and explored alternative methods to tweak $d$. Specifically, instead of attempting to recover the full dimension $d$, our strategy focuses on recovering a dimension $d'$ where $d' \leq d$. This approach sets the subweights $(d:, d':)$ of $W_1$ to zero. Then the gradient $g_j$ in Equation 18 remains functional but is exclusively tied to the subweights $(:, :d')$ of $W_1$. As a result, we embrace a more directed and efficient methodology by centering our reconstruction on the feature subset $(B, d')$.

**Challenges in Order Recovery of Features :** When applying tensor decomposition techniques to retrieve features from $\hat{T}$, a significant issue arises when the batch size exceeds one: the exact order of the recovered features remains uncertain. Under adversarial conditions, one might try every conceivable permutation as a reference. However, we simplify the procedure by sequentially comparing each recovered feature to the actual input features with cosine similarity until the best order is discerned. In certain cases, a single recovered feature displayed a notably high cosine similarity with multiple actual inputs simultaneously. Interestingly, although a 1-m greedy relationship might exhibit a high correlation, it did not exceed the attack performance of a straightforward 1-1 match in the final outcome. Consequently, we adopted the 1-1 relationship to achieve the best attack result.

### 4.3 FEATURE MATCH

Following Balunovic et al. (2022), we have segmented our entire text retrieval process into three phases: Initialization, Optimization, and Token Swap. In the initialization and token swap stages, we aim to leverage certain metrics to identify optimal starting or intermediary points for the subsequent optimization phase. This method is also commonly recognized as discrete optimization. In this setting, we've chosen a mix of metrics to guide the choice, including gradient match loss and perplexity obtained from pre-trained language models. More details can be found in Balunovic et al. (2022). In the optimization stage, we propose to optimize the embeddings derived from input IDs and the features directed into the Pooler layer simultaneously. We use gradient match loss and cosine distance between the input of the Pooler layer with the recovered intermediate features from Section 4.2 to guide the optimization. Moreover, we oscillate between continuous and discrete optimization phases to bolster the final attack performance.

### 5 EXPERIMENTS

#### 5.1 SET UP

**Datasets:** Following previous work Balunovic et al. (2022), our experimental design incorporates three binary text classification datasets to ensure a comprehensive evaluation. Specifically, we uti-

lize **CoLA** and **SST-2** from the **GLUE** benchmark (Warstadt et al., 2018; Socher et al., 2013; Wang et al., 2019), with their sequences predominantly ranging between 5-9 and 3-13 words, respectively. Additionally, the **RottenTomatoes** dataset presents a more complex scenario with sequence lengths oscillating between 14 and 27 words (Pang & Lee, 2005). You may find more details about datasets in the appendix. Within the scope of our experiments, we utilize a subset of 100 randomly selected sequences from the training sets of these datasets as our evaluation benchmark, a method also endorsed by Balunovic et al. (2022).

**Models:** We conduct experiments primarily on the **BERT$_{BASE}$** (Devlin et al., 2018) architecture. Consistent with Balunovic et al. (2022), we use models that have undergone fine-tuning for downstream tasks over two epochs. To ensure a fair comparison, we even adopt the same fine-tuned models from Balunovic et al. (2022). As for the auxiliary language model employed to extract prior knowledge, we choose GPT-2 (Radford et al., 2019), a choice also used by Balunovic et al. (2022).

**Metrics:** Following Deng et al. (2021) and Balunovic et al. (2022), we evaluate attack performance using the ROUGE metric suite (Lin, 2004). Specifically, we present the collective F-scores for **ROUGE-1**, **ROUGE-2**, and **ROUGE-L**. These metrics respectively assess the retrieval of unigrams, bigrams, and the proportion of the longest continuous matching subsequence relative to the entire sequence's length. We omit all padding tokens in the reconstruction and evaluation phases.

**Baselines:** We benchmark our approach against three key baselines: **DLG**, **TAG**, and **LAMP**. Among them, LAMP represents the state-of-the-art. We employ the open-sourced implementation from LAMP, which encompasses the implementations for all three baselines (Deng et al., 2021; Zhu et al., 2019; Balunovic et al., 2022). Following previous work, we assume the lengths of sequences are known for both baselines and our attacks, as an adversary can run the attack for all possible lengths (Balunovic et al., 2022).

**Implementation:** Our method is implemented based on LAMP's framework, utilizing the exact same datasets, evaluation metrics, and similar models. To ensure a fair comparison, we standardized the experimental conditions and settings when comparing our approach with baselines, particularly the state-of-the-art LAMP. We adopt all of LAMP's hyperparameters, including the optimizer, learning rate, learning rate schedule, regularization coefficient, optimization steps, and random initialization numbers. For hyperparameters unique to our method, we made selections using a grid search on BERT$_{BASE}$ and shared them in different settings (LAMP also adopts this strategy). It's also important to note that all our experiment results are averaged over five different random seeds.

## 5.2 RESULTS AND ANALYSIS

We present experimental results in Table 1. These findings clearly demonstrate that our approach outperforms all baselines (DLG, TAG, and LAMP) across various datasets and batch sizes. There's an average improvement of up to **9.3%** for ROUGE-1, **6%** for ROUGE-2, and **7%** for ROUGE-L.

Examining the impact of batch size variations, we notice that launching an attack becomes more challenging as the batch size increases. All attack methods, including ours, exhibit a decline in attack performance. However, our method brings a more noticeable improvement at batch sizes 2 and 4, surpassing its efficacy at batch sizes 1 and 8. We posit that for a batch size of 1, where the gradient is only averaged solely over tokens, the benefit of incorporating the feature information is less evident because the gradient information still plays a leading role in the optimization process. For a batch size of 8, the improvement scale is also not pronounced, we explore the background reason in Section 5.3.

Turning our attention to variations in sequence length across datasets, we notice a clear trend: as sequences get longer, the benefit from intermediate features at a batch size of 1 becomes more pronounced. Specifically, for the CoLA dataset with token counts between 5-9, we see an average improvement in ROUGE metrics of **3%**. This improvement grows to **5%** for the SST-2 dataset with token counts from 2 to 13. For the Rotten Tomatoes dataset, which features even longer sequences with token counts ranging from 14 to 27, the average ROUGE metric improvement further increases to **8%**. This suggests a correlation between sequence length and the extent of improvement observed. However, when the batch size exceeds one, the benefits observed for these three datasets are consistently notable. Recall that gradient averaging occurs only over tokens at a batch size of 1, it implies that with longer sentences, the gradient information becomes less effective, leading to

Table 1: Text privacy attack on BERT$_{BASE}$ with Different Batch Sizes and Datasets. R-1, R-2, and R-L, denote ROUGE-1, ROUGE-2, and ROUGE-L scores respectively.

| Method | B=1 | | | B=2 | | | B=4 | | | B=8 | | |
|---|---|---|---|---|---|---|---|---|---|---|---|---|
| | R-1 | R-2 | R-L | R-1 | R-2 | R-L | R-1 | R-2 | R-L | R-1 | R-2 | R-L |
| **CoLA** | | | | | | | | | | | | |
| DLG | 59.3 | 7.7 | 46.2 | 36.9 | 2.6 | 31.4 | 35.3 | 1.4 | 31.9 | 16.5 | 0.8 | 7.9 |
| TAG | 78.9 | 10.2 | 53.3 | 45.6 | 4.6 | 36.9 | 35.3 | 1.6 | 31.3 | 33.3 | 1.6 | 30.4 |
| LAMP $_{COS}$ | 84.8 | 46.2 | 73.1 | 57.2 | 21.9 | 49.8 | 40.4 | 6.4 | 36.2 | 36.4 | 5.1 | 34.4 |
| **Ours** $_{SELU}$ | **86.6** | **51.5** | **76.7** | **69.5** | **31.2** | **60.6** | **50.5** | **11.8** | **43.9** | **40.8** | **8.3** | **38.1** |
| **Ours** $_{x^3+x^2}$ | 84.6 | 45.2 | 72.4 | 57.3 | 19.2 | 49.8 | 43.9 | 11.4 | 40.1 | 37.8 | 5.9 | 34.8 |
| **SST-2** | | | | | | | | | | | | |
| DLG | 57.7 | 11.7 | 48.2 | 39.1 | 7.6 | 37.2 | 38.7 | 6.5 | 36.4 | 36.6 | 4.7 | 35.5 |
| TAG | 71.8 | 16.1 | 54.4 | 46.1 | 10.9 | 41.6 | 44.5 | 9.1 | 40.1 | 41.4 | 6.7 | 38.9 |
| LAMP $_{COS}$ | 87.7 | 54.1 | 76.4 | 59.6 | 26.5 | 53.8 | 48.9 | 17.1 | 45.4 | 39.7 | 10.0 | 38.2 |
| **Ours** $_{SELU}$ | 90.3 | 59.0 | 78.2 | 71.0 | 35.3 | 63.4 | 58.6 | **26.3** | 54.2 | 45.4 | 11.5 | 43.2 |
| **Ours** $_{x^3+x^2}$ | **93.1** | **61.6** | **81.5** | **78.3** | **40.9** | **67.9** | **60.6** | 23.1 | **54.9** | **49.5** | **16.5** | **47.3** |
| **Rotten Tomatoes** | | | | | | | | | | | | |
| DLG | 20.1 | 0.4 | 15.2 | 18.9 | 0.6 | 15.4 | 18.7 | 0.4 | 15.7 | 20.0 | 0.3 | 16.9 |
| TAG | 31.7 | 2.5 | 20.1 | 26.9 | 1.0 | 19.1 | 27.9 | 0.9 | 20.2 | 22.6 | 0.8 | 18.5 |
| LAMP $_{COS}$ | 63.4 | 13.8 | 42.6 | 38.4 | 6.4 | 28.8 | 24.6 | 2.3 | 20.0 | 20.7 | 0.7 | 17.7 |
| **Ours** $_{SELU}$ | 71.9 | 19.2 | 48.7 | **48.1** | **8.2** | **34.2** | **33.0** | **4.23** | **25.3** | **24.6** | **2.0** | **20.6** |
| **Ours** $_{x^3+x^2}$ | **72.2** | **21.0** | **49.3** | 44.6 | 7.0 | 31.8 | 29.9 | 3.5 | 24.3 | 23.6 | 1.7 | 19.8 |

greater benefits from intermediate feature supervision signals. When batch sizes are larger than 1, averaging happens over tokens and sentences simultaneously. This broadened scope results in our method consistently yielding pronounced benefits across sequences with different lengths. Our findings further reinforce the idea that relying exclusively on gradient information diminishes efficacy with larger batch sizes and longer sequences.

Additionally, with the inclusion of feature information as a supervision signal, our method can recover not only a greater number of tokens but also more accurate token orderings. In comparison to other baselines, we can recover longer text sequences. The improvement in ROUGE-2 and ROUGE-L metrics supports these observations.

Table 2: Influence of cosine distance in different text retrieval phases on BERT$_{BASE}$ and SST-2 dataset

| Phase | R-1 | R-2 | R-L |
|---|---|---|---|
| Batch Size=1 | | | |
| Non-use (LAMP) | 87.7 | 54.1 | 76.4 |
| Only Discrete | 92.5 | 59.3 | 79.9 |
| Only Continuous | **93.1** | **61.6** | **81.5** |
| Both | 90.0 | 53.9 | 76.8 |
| Batch Size=4 | | | |
| Non-use (LAMP) | 48.9 | 17.1 | 45.4 |
| Only Discrete | 57.9 | 23.4 | 52.3 |
| Only Continuous | 60.6 | 23.1 | 54.9 |
| Both | **61.7** | 23.0 | **55.7** |

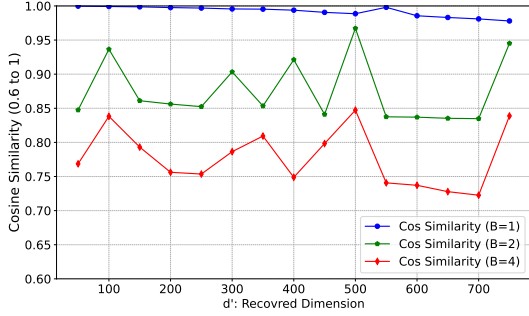

Figure 2: Cosine similarity between recovered features and ground Truth on BERT$_{BASE}$ across varying dimensions (50-750 in 50-step intervals) and batch sizes (1, 2, 4)

## 5.3 DISCUSSION

**Impact of Recovery Dimension:** In Section 4.2, we propose fixing $m$ and adjusting $d'$ to identify the optimal mapping for $d'$ (where $d' < d$) and $m$. Accordingly, we conduct experiments using

$BERT_{BASE}$ with various batch sizes to investigate the quality of the recovered intermediate features by calculating their cosine similarity with the ground truth. The results are illustrated in Figure 2. Our findings suggest that when the batch size is 1, the recovered quality gradually degrades as the recovery dimension $d'$ increases, yet it remains as high as 0.99 across all configurations. However, this pattern does not hold when the batch size exceeds 1. We also observed that the recovered quality consistently declines as the batch size increases. We hypothesize that multiple inputs might exhibit some undisclosed dependencies, particularly features within the deeper layers of language models, thereby affecting the efficacy of tensor decomposition. For simplicity, we set $d' = 100$ across all experiments. However, under adversarial conditions, attackers might experiment with various $d'$ settings to enhance their attack performance.

**Impact of Feature Match in Different Optimization Phase:** In Section 4.3, we propose a novel optimization objective: the cosine distance between the input of the Pooler layer and the recovered intermediate features from Section 4.2. It's worth noting that we can also apply this distance as a new metric like gradient match loss in the discrete optimization stage to select the best starting or intermediary points for the subsequent training phase. Therefore, we add the new metric to the discrete and continuous optimization phases separately to observe its impact on the final attack performance. The results are illustrated in Table 2. Notably, our introduced metric has a positive effect on both phases. However, when the new metric is used in discrete and continuous optimization together, the results are not always two-win.

Table 3: Text privacy attack on RoBERTa $_{BASE}$. R-1, R-2, and R-L are same within Table 1. Cos$_S$ indicates the average cosine similarity between references and recovered samples.

| Dataset | Method | R-1 | R-2 | R-L | Cos$_S$ | Recovered Samples |
|---|---|---|---|---|---|---|
| CoLA | \multicolumn{6}{c}{reference sample: The box contains the ball} | | | | | |
| | LAMP | 15.5 | 2.6 | 14.4 | 0.36 | likeTHETw box contains divPORa |
| | **Ours** | **17.4** | **3.8** | **15.9** | **0.41** | like Mess box contains contains balls |
| SST2 | \multicolumn{6}{c}{reference sample: slightly disappointed} | | | | | |
| | LAMP | **20.1** | **2.2** | 15.9 | 0.56 | likesmlightly disappointed a |
| | **Ours** | 19.7 | 2.1 | **16.8** | **0.59** | like lightly disappointed a |
| Toma | \multicolumn{6}{c}{reference sample: vaguely interesting, but it's just too too much} | | | | | |
| | LAMP | 19.9 | 1.6 | 15.1 | 0.48 | vagueLY', interestingtooMuchbuttoojusta |
| | **Ours** | **21.5** | **1.8** | **16.0** | **0.51** | vagueLY, interestingBut seemsMuch Toolaughs |

**Impact on Other Models:** To demonstrate the effectiveness of our attack method on various model architectures, we also apply our method on the RoBERTa (Liu et al., 2019). While RoBERTa shares similarities with BERT, it distinguishes itself through unique training configurations and datasets. Notably, unlike $BERT_{BASE}$, RoBERTa does not have a Pooler layer. Instead, it employs a classifier composed of two linear layers in the head. In our experiments, we treat the first layer as an analogous Pooler layer and endeavor to reconstruct its input. All the models used in this experiment are from Hugging Face, contributed by TextAttack. As for the auxiliary model, we employ RoBERTa itself due to a specific challenge: we can't locate another generative model using the same tokenizer with RoBERTa. However, it's essential to note that we use the exact same settings for baselines and our method. We present the experiment results in Table 3. While the overall attack performance significantly decreases due to the auxiliary masked language model, our approach still outperforms the baseline. Furthermore, in numerous instances (as illustrated in Table 3), our method appears to restore the essence of the reference sample almost flawlessly. However, due to the limitation of traditional evaluation metrics, they may have equal or even worse evaluation numbers than some obvious bad recovery. Therefore, we propose to use the cosine similarity between the embeddings of reference and recovery generated by SBERT (Reimers & Gurevych, 2019).

## 6 CONCLUSION

This paper presents a novel method for text privacy attacks that is difficult to detect and defend. Instead of solely relying on traditional gradients and prior knowledge, our approach incorporates unique feature-level information. Comprehensive empirical studies across various model architectures, datasets, and batch sizes affirm the effectiveness of our method.

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

# A  APPENDIX-A

## A.1  DETAILED DISCUSSION ON THREAT MODEL AND ATTACK METHODOLOGY

In Section 2, following previous studies, we categorize private textual data attacks into two main types: **Malicious Attacks** and **Eavesdropping Attacks**, the latter also referred to as *honest-but-curious* Attacks (Gupta et al., 2022). However, this classification oversimplifies the issue, missing many nuanced intermediary states. This section aims to provide a clearer understanding of our threat model and the specifics of our attack methodology, highlighting how our approach differs from previous attacks and its positioning in the context of current research.

### A.1.1  COMPARISION WITH MALICIOUS ATTACK

Conventionally, Malicious Attack involves issuing malicious server states (parameters) or even altering model architectures (Fowl et al., 2021; 2022; Boenisch et al., 2023). Under such a definition, when an attack occurs, the federated learning protocol is compromised, as the generated gradient information is valueless for minimizing the loss. However, in our settings, we do not change the inherent purpose of the gradients and do not affect the original training process of the model. From this perspective, our method does not align with a typical Malicious Attack.

Furthermore, our modifications are limited exclusively to the Pooler layer, which includes expanding the model's width and altering the activation function. These changes are relatively minor, especially when compared to the extensive alterations often observed in Malicious Attack. Previous research has often entailed extensive alterations to model parameters, including multi-layer adjustments and even the introduction of entirely new modules. These approaches are significantly more aggressive compared to our modifications Fowl et al. (2021; 2022); Boenisch et al. (2023).

**Wider Pooler Layer:**  Applying our method to the Pooler layer, while requiring a wide setting ($d_{out} \gg d_{in}$), depends on the recovery needs. For a small input portion ($x'$ with $d'_{in} \ll d_{in}$), the required width reduces significantly, as detailed in Section 4.2: **Flexibility of Recovered Dimension**. With the growth of Large Language Models like GPT-3 (Brown et al., 2020), which already exceeds the necessary dimensions (12288 for $d'_{in} = 50$), this width requirement becomes less vital. Moreover, our attack focuses on wide models, aligning with previous studies (Geiping et al., 2020; Fowl et al., 2022), revealing more risk in our settings.

**Activation Functions Change:**  We have identified risks related to using activation functions such as Selu, whose higher-order derivatives are neither odd nor even. Our ongoing research extends to activation functions with strictly odd or even derivatives, such as Tanh or Relu. While many previous studies have focused on specific activation functions, we view this discovery as a contribution, not a drawback (Zhu et al., 2019; Fowl et al., 2021).

### A.1.2  COMPARISION WITH EVASDROPING ATTACK

Previous studies like Balunovic et al. (2022) can extract private text information using only optimization-based techniques and prior knowledge without any model modifications. However, this is a self-imposed limitation. In reality, servers can undertake actions well beyond this scope. However, such actions are mostly easy-detectable, our method achieves less detectability while providing additional optimization signals, marking a significant innovation.

Compared to Zhu et al. (2019); Deng et al. (2021); Balunovic et al. (2022), our method does involve some modifications to the model. However, we propose that *honest* should not be narrowly defined as the server's absolute fidelity to clients. Instead, *honest* should be understood as the server's adherence to its primary role of obtaining effective gradients and minimizing the training loss. By this definition, our approach falls under the *honest-but-curious* category.

### A.1.3  CLARIFYING THE METHOD'S POSITIONING

Our approach, distinct from the above Malicious or Eavesdropping Attacks, offers more effective outcomes without violating the fundamental objectives of federated learning. Our focus has primarily been on the optimization aspect due to these considerations. We clarify the **threat model** of our method as follows:

- We do not fine-tune the model's **token and positional embeddings**. The gradients for these embeddings are non-zero for words in the client's training data, enabling easy recovery of client sequences.

- The server operates under a ***honest-but-curious*** model, aiming to learn maximally from gradients without deviating from the fundamental objectives of federated learning, like effective gradient aggregation and model loss minimization.

To enhance understanding, a comparative analysis of different attack categories, including Malicious Attacks, Eavesdropping Attacks, and Honest-but-Curious Server attacks, are presented in Table 4, with references to relevant studies.

Table 4: Comparative analysis of different attack categories on textual data. Please note that we do not consider the attack methods that require fine-tuning the token and positional embeddings. (The term ***honest*** signifies the server's intention to extract client sequences or information from gradients, while still adhering to the core principles of federated learning, such as efficient gradient aggregation and minimizing model loss).

| Attack Category | Malicious | Eavesdropping | Honest-but-Curious (Ours) |
|---|---|---|---|
| Degree of Model Modification | Heavy | None | Light |
| Model Modification | Multiple Layers | None | Single Layer |
| Insertion of New Module | Yes / None | None | None |
| Attack Constraint | Heavy | Light | Moderate |
| FL Protocol Broken | Yes | None | None |
| Attack Effectiveness | Strong | Weak | Moderate |
| Previous Studies | Fowl et al. (2021) | Balunovic et al. (2022) | Ours |
| | Fowl et al. (2022) | Zhu et al. (2019) | |
| | Boenisch et al. (2023) | Deng et al. (2021) | |

## A.2 SPECIFICITY OF ATTACK SCENARIOS

Research in privacy attacks holds a unique importance in practice. This is because every discovery in this area, even with certain constraints, can cause tremendous destruction once it happens. The attacker will try their best to satisfy the constraints because of the hidden massive profit. Therefore, the risks uncovered by these studies are often difficult to measure, making them crucial for understanding security vulnerabilities.

# B APPENDIX-B

## B.1 EXTEND TO CROSS ENTROPY LOSS

Wang et al. (2023) grounded their research on the assumption that the loss function of the neural network is Mean Square Error (MSE). Building upon this foundation, we extend the method to the scenario of classification tasks utilizing Cross-Entropy Loss (CEL). In the classification context, the gradient of $g_j$ is calculated for all class outputs. While a straightforward approach might only random choose the gradient for a single class to satisfy the equation 18, we chose a more holistic method, leveraging the gradient of the pooler layer to compute $\hat{T}$ rather than the classifier layer. Based on this methodology, the gradient of $w_j$ we derived is as follows:

$$\hat{g}_j = \nabla_{w_j} L(\Theta) = \sum_{i=1}^{B} r_i a_j \sigma' \left( w_j^\top x_i \right) x_i \tag{7}$$

Let $a_j = \frac{1}{m}, \forall j \in [m]$ and $w_j \in N(0, 1)$, by Stein's lemma, we have:

$$T_1 = \sum_{i=1}^{m} \hat{g}_j H_2(w_j) \tag{8}$$

$$= \frac{1}{m} \sum_{i=1}^{B} r_i^* x_i \otimes \left[ \sum_{j=1}^{m} \sigma' \left( w_j^\top x_i \right) \left( w_j \otimes w_j - I \right) \right] \tag{9}$$

$$\approx \sum_{i=1}^{B} r_i^* x_i \otimes \mathbb{E} \left[ \sigma' \left( w_j^\top x_i \right) \left( w_j \otimes w_j - I \right) \right] \tag{10}$$

$$= \sum_{i=1}^{B} r_i^* \mathbb{E} \left[ \sigma^{(3)}(w^\top x_i) \right] x_i^{\otimes 3} \tag{11}$$

$$= T \tag{12}$$

By defining the tensors $T2$ and $T3$ such that: $T2(i,j,k) = T1(k,i,j)$ and $T3(i,j,k) = T1(j,k,i)$, we can deduce: $\hat{T} = \frac{T + T_3^2 + T_3}{3} \approx T$. This computation results in $\hat{T}$ being symmetric. Wang et al. (2023) even observed that this method offers a more precise estimation when attempting to recover features. We also adopt this strategy in all our experiments.

## B.2   IMPACT OF ACTIVATION FUNCTION

When applying the two-layer-neural-network-based reconstruction method to the Pooler layer of language models, we also substitute the original Tanh activation function with the ReLU. However, the third-order derivative of the ReLU function is odd, leading to zero expectation $\mathbb{E}_{Z \sim N(0,1)}[\sigma^{(3)}(Z)] = 0$. This property of the ReLU renders it unstable for third-order tensor decomposition. To address this challenge, we follow the approach proposed by Wang et al. (2023), instead of using a third-order Hermite function to estimate $T$, we use a fourth-order function. The estimation is represented as:

$$\hat{T} := \frac{1}{m} \sum_{j=1}^{m} g_j(w_j) H_4(w_j)(I, I, I, a) \tag{13}$$

where $a$ is a unit vector, pointing in a specific direction in space. However, the result of the experiment is not ideal even compared with baselines, which means we need to find a more practical method to resolve this problem.

## B.3   IMPACT OF DATA DEPENDENCE

We made a noteworthy observation during our implementation of the two-layer-neural-network-based reconstruction technique. When the batch size goes beyond a single data point, ensuring the independence of features across various data points becomes crucial. However, there's an inherent challenge in achieving this. Delving deeper into the language model, particularly close to the Pooler layer, we find that dominant features are those closely aligned with the downstream task. Using sentiment analysis as an example, features directed to the Pooler layer somewhat have characteristics that describe similar emotions. Unfortunately, this similarity can degrade the quality of the features we are trying to recover. As a result, the reliability of these recovered features might be diminished when they are used as ground truth during optimization.

Wang et al. (2023)'s analysis also underscores this puzzle: the reconstruction quality is closely tied to the condition number, defined by the data matrix's smallest singular value. To elaborate further, if a sample is heavily influenced by or dependent on other samples (like two sentences mirroring each other or belonging to identical classes), the assurance of accurate recovery falters. This decline is attributed to the inherent limitation of tensor decomposition when faced with almost identical data. For instance, with two strikingly similar sentences, tensor decomposition might only be able to discern the collective span of the sentences, failing to distinguish between them. Resorting to feature matching in such scenarios would invariably perform negatively.

## C APPENDIX-C

### C.1 CLARIFICATION ON TWO-LAYER-NEURAL-NETWORK-BASED RECONSTRUCTION

Consider a two-layer neural network: $f(x; \Theta) = \sum_{j=1}^{m} a_j \sigma(w_j \cdot x)$, with parameters defined as $\Theta = (a_1, ..., a_m, w_1, ..., w_m)$. Here, $m$ represents the hidden dimension. The objective function is represented as: $L(\Theta) = \sum_{i=1}^{B}(y_i - f(x_i; \Theta))^2$. A notable finding is that the gradient for $a_j$ is solely influenced by $w_j$, making it independent from other parameters. This gradient is represented as:

$$g_j := \nabla_{a_j} L(\Theta) = \sum_{i=1}^{B} r_i \sigma\left(w_j^\mathsf{T} x_i\right) \tag{14}$$

where the residual $r_i$ is given by $r_i = f(x_i; \Theta) - y_i$. For wide neural networks with random initialization from a standard normal distribution, the residuals $r_i$ concentrate to a constant, $r_i^*$. By set $g_{(w)} := \sum_{i=1}^{B} r_i^* \sigma(w^\mathsf{T} x_i)$, $g_j$ can be expressed as $g_j = g(w_j) + \epsilon$, where $\epsilon$ represents noise. This is to say by setting different $w$ we are able to observe a noisy version of $g(w)$, where we have the first order derivative of $g(w)$:

$$\nabla g(w) = \sum_{i=1}^{B} r_i^* \sigma'(w^\mathsf{T} x_i) x_i \tag{15}$$

Similarly, we have the second and third derivations of $g(w)$:

$$\nabla^2 g(w) = \sum_{i=1}^{B} r_i^* \sigma''(w^\mathsf{T} x_i) x_i x_i^\mathsf{T} \tag{16}$$

$$\nabla^3 g(w) = \sum_{i=1}^{B} r_i^* \sigma^{(3)}(w^\mathsf{T} x_i) x_i^{\otimes 3} \tag{17}$$

Here, $x_i^{\otimes 3}$ signifies the tensor product of vector $x_i$ with itself three times. Given $E_W \nabla^p g(W)$, where $p = 1, 2, 3$, we are able to recover the reweighted sum for $x_i^{\otimes p}$. Especially when $p = 3$, the third order tensor $E_w \nabla^3 g(W)$ has a unique tensor decomposition which will identify $\{x_i\}_{i=1}^{B}$ when they are independent. Wang et al. (2023) further take use of Stein's Lemma, expressed as: $\mathbb{E}[g(X)H_p(X)] = \mathbb{E}[g^{(p)}(X)]$ to approximate $E_W \nabla^3 g(W)$ as:

$$T = \mathbb{E}_W[\nabla_W^3 g(W)] = \mathbb{E}_{W \sim N(0,I)}[g(W)H_3(W)] \approx \frac{1}{m} \sum_{j=1}^{m} g(w_j) H_3(w_j) = \hat{T} \tag{18}$$

Where $H_3(w_j)$ is the p-th Hermite function of $w_j$. In this way, we have a very close estimation $\hat{T} \approx T$, and take use of the technique of tensor decomposition, we can recover the unique $x_i$. **However, we want to reinforce the directional component of the feature in the recovered information. Yet, recovering the magnitude information of the feature remains a challenging task**. For more details, please refer to the paper Wang et al. (2023).

### C.2 INTUITION OF INTERMEDIATE FEATURES

Two previous works that utilize intermediate features to enhance privacy and adversary attacks also share a similar intuition with ours (Huang et al., 2019; Kariyappa et al., 2023). However, Huang et al. (2019) focuses on a completely different attack scenario and objective, with different constraints and limitations in the community. In contrast, Kariyappa et al. (2023) is a recent work that concentrates on image data recovery, employing intermediate features in the context of federated learning. Considering that attacking private text data presents unique challenges compared to image data, and that our methods differ from these studies, our novelty and contribution remain distinct. We are encouraged to find that our approach shares similar intuition with these two studies.

C.3 DATASETS

**CoLA**: The CoLA (Corpus of Linguistic Acceptability) dataset is a seminal resource for evaluating the grammatical acceptability of machine learning models in natural language processing. Consisting of approximately 10,657 English sentences, these annotations are derived from various linguistic literature sources and original contributions. The sentences are categorized based on their grammatical acceptability. Spanning a comprehensive range of linguistic phenomena, CoLA provides a robust benchmark for tasks requiring sentence-level acceptability judgments. Its diverse set of grammatical structures challenges models to demonstrate both depth and breadth in linguistic understanding, making it a popular choice in the field.

**SST-2**: The SST-2 (Stanford Sentiment Treebank Version 2) dataset is a widely recognized benchmark for sentiment analysis tasks in natural language processing. Originating from the Stanford NLP Group, this dataset contains around 67,000 English sentences, drawn from movie reviews, annotated for their sentiment polarity. Unlike its predecessor which had fine-grained sentiment labels, SST-2 has been simplified to a binary classification task, where sentences are labeled as either positive or negative. This dataset not only provides sentence-level annotations but also contains a unique feature: a parsed syntactic tree for each sentence. By leveraging both sentiment annotations and syntactic information, we can investigate various dimensions of sentiment understanding and representation in machine learning models.

**Rotten Tomatoes**: The Rotten Tomatoes dataset is a compilation of movie reviews sourced from the Rotten Tomatoes website. This dataset has been instrumental in sentiment analysis research. In its various versions, the most notable being SST-2, the dataset consists of sentences from these reviews, annotated for their sentiment polarity. These sentences are labeled either as positive or negative, making it a binary classification challenge. The dataset's value lies in its representation of real-world opinions, rich in diverse sentiment expressions, and has been a cornerstone for evaluating the performance of natural language processing models in sentiment classification tasks.

| Reference | Recovery |
|---|---|
| slightly disappointed | slightly disappointed |
| splendidly | splendidly |
| gaining much momentum | gaining much momentum |
| flawless film | flawless film |
| tiresomely | tiresomely |
| enjoyable ease | ease enjoyable |
| grayish | grayish |
| no cute factor here ... not that i mind ugly ; the problem is he has no character , loveable or otherwise . | he no problem is here i really love cute, not ugly the mind or no character ; the loveable love factor cute has. |
| of softheaded metaphysical claptrap | softhead of metaphysical clap claptrap |
| ably balances real-time rhythms with propulsive incident . | time ably balances incident with real incident.ulsive rhythms. |
| was being attempted here that stubbornly refused to gel | here was attempted stubbornly that being refused to gel |
| that will be seen to better advantage on cable , especially considering its barely | , that better to barely advantage will be seen on cable considering its advantage |
| point at things that explode into flame | point things flame that explode into explode |
| undeniably intriguing film | undeniably intriguing film |
| efficient , suitably anonymous chiller . | efficient, suitably anonymous chiller shady |
| all of this , and more | this and all this more, |
| want to think too much about what s going on | think want to think too much about what s going on |
| invigorating | invigorating |
| to infamy | to infamy |
| the perverse pleasure | the perverse pleasure |
| the way this all works out makes the women look more like stereotypical caretakers and moral teachers , instead of serious athletes . | the stereotypical this way all works out ( the more like oxygenmissible caretaker makes teachers of athletes instead look moral. women instead |
| a successful adaptation and an enjoyable film in its own right | a successful and enjoyable film adaptation right in its own right |
| while some will object to the idea of a vietnam picture with such a rah-rah , patriotic tone , soldiers ultimately achieves its main strategic objective : dramatizing the human cost of the conflict that came to define a generation . | will achieve object main while idea conflict drama with the such tone a political picture cost : vietnam thetih ra, vietnam insulted achieves objective objective, some patriotic dramazing a tone of soldiers generation that strategic its drama ultimately generation to define. |
| taken outside the context of the current political climate ( see : terrorists are more evil than ever ! ) | the climate terrorists than outside the context of current political climate ( see : are evil ever taken! ) |
| strange and beautiful film | strange and beautiful film |
| this ) meandering and pointless french coming-of-age import from writer-director anne-sophie birot | this meander pointless director - anne french - coming from pointless importing of writer ) and ageing - -rot |
| are so generic | are so generic |
| for only 71 minutes | for 71 minutes only |
| i also believe that resident evil is not it . | it is also i not.. believe resident evil |
| fizzability | fizzability |

| Reference | Recovery |
|---|---|
| a better vehicle | a better vehicle |
| pull together easily accessible stories that resonate with profundity | hand together stories resonate with pullclundity easily accessible |
| higher | higher |
| build in the mind of the viewer and take on extreme urgency . | build urgency in the extreme of viewer urgency and take on mind. |
| we ve seen it all before in one form or another , but director hoffman , with great help from kevin kline , makes us care about this latest reincarnation of the world s greatest teacher . | thesegreatest of form seen beforeall reinnationdirector we, directorstand wele great hoffman in ve latest makes us help teacher care about greatestnation in this thelancenation, but one of |
| s horribly wrong | shorribly wrong |
| eccentric and | eccentric and |
| scare | scare |
| finds one of our most conservative and hide-bound movie-making traditions and gives it new texture , new relevance , new reality . | gives our finds new finds, conservative new-bound movie making traditions - and reality texture it hide. reality texture and one movie relevance |
| pummel us with phony imagery or music | imagery pummel us or phony with music |
| consistently sensitive | consistently sensitive |
| the project s filmmakers forgot to include anything even halfway scary as they poorly rejigger fatal attraction into a high school setting . | s scary filmmakers forgot anything forgot to include even halfway fatal attraction as they poorlyjigger regger into high school scary project setting |
| narcissistic | narcissistic |
| has been lost in the translation ... another routine hollywood frightfest in which the slack execution italicizes the absurdity of the premise . | slack has the includesity in the executionalic translation. another frightfest. the absurd premise which lost, it routineizes the premise of hollywood. |
| – bowel movements than this long-on-the-shelf , point-and-shoot exercise in gimmicky crime drama | movements - - than long - shoot - - this exercise, and this - the bowel shelf - on gimmick in crime drama point |
| visually striking and slickly staged | visually striking and slickly staged |
| downright transparent | downright transparent |
| rotting underbelly | underbelly rotting |
| could possibly be more contemptuous of the single female population . | could possibly be more contemptuous of the single female population. |

| Reference | Recovery |
|---|---|
| what the english call ' too clever by half | what ' call call by clever english too half |
| sucks , but has a funny moment or two . | has funny sucks but moment or two funny sucks. |
| trailer-trash | trash trailer - |
| flinching | flinching |
| hot topics | hot topics |
| settles too easily | settles too easily |
| films which will cause loads of irreparable damage that years and years of costly analysis could never fix | films which will cause loads ofparable damage that years and years of costly analysis irre could never fix |

| Reference | Recovery |
|---|---|
| wears | wears |
| is an inspirational love story , capturing the innocence and idealism of that first encounter | innocence is an inspirational story capturing the idealism of first encounter, and love that |
| has the charisma of a young woman who knows how to hold the screen | has the the thea of char young who knows how hold of screen womanism |
| circuit is the awkwardly paced soap opera-ish story . | h - is awkwardly paced circuit story is the soap opera story |
| , beautiful scene | beautiful scene, |
| grace to call for prevention rather than to place blame , making it one of the best war movies ever made | to call for prevention rather than to place blame, grace making it one of the best war movies ever made |
| looking for a return ticket | looking for a return ticket |
| the strange horror | the strange horror |
| , joyous romp of a film . | , a joyous romp of film. |
| a longtime tolkien fan | a longtime tolkien fan |
| heartwarming , nonjudgmental kind | heartwarming, nonmingjugmental kind |
| uncouth , incomprehensible , vicious and absurd | absurdhensible, uncouth, vicious and incompmbled |
| a real winner – smart , funny , subtle , and resonant . | a winner. resonant and funny - ami subtle, smart, real res |
| gets clunky on the screen | gets on screenunk clunky |
| there s not a single jump-in-your-seat moment and | there s not a single jump and seat in your seat - - - moment |
| has a tougher time balancing its violence with kafka-inspired philosophy | acter has a tough time balancing itsfka philosophy with violence - inspired |
| bad filmmaking | bad filmmaking |
| share | share |
| this excursion into the epicenter of percolating mental instability is not easily dismissed or forgotten . | this excursionenter is the mentalenter into instability or iserving easily dismissed or not easily forgotten. |
| s as if allen , at 66 , has stopped challenging himself . | as if regarding sums, allen has stopped s 66, challenging himself. |
| is its make-believe promise of life that soars above the material realm | its promise that life is promiseence make soars above the material realm - |
| exit the theater | exit the theater |
| is fascinating | fascinating is |
| wise , wizened | wise, wizened |
| is not the most impressive player | is not the most impressive player |
| it s undone by a sloppy script | its undone by a sloppy script |
| know what it wants to be when it grows up | know what grows up when it wants it to be |
| people have lost the ability to think | people have lost the ability to think |
| unfortunately , it s also not very good . | . very, unfortunately it also s not very good |
| clarity and emotional | and emotional clarity |
| propulsive | propulsive |
| p.t. anderson understands the grandness of romance and how love is the great equalizer that can calm us of our daily ills and bring out joys in our lives that we never knew were possible . | l of will understands joy is our romance. daily we ill of how of t a grand anderson. the anderson romanceing calms never at us lives guest bearings daily and ofness of coulds p the grand. |

| Reference | Recovery |
|---|---|
| tactic to cover up the fact that the picture is constructed around a core of flimsy – or , worse yet , nonexistent – ideas | tactic to cover up the fact picture the core or the coreim constructed,' - none worse yet - - aroundum orstensyim. and central ideas |
| how ridiculous and money-oriented | how ridiculous and - money oriented |
| muy loco , but no more ridiculous | muy loco, but no more ridiculous |
| deceit | deceit |
| in its understanding , often funny way | understanding in its often funny way, |
| a caper that s neither original nor terribly funny | s that original a caper neither original nor terribly funny |
| ( denis ) story becomes a hopeless , unsatisfying muddle | denis use ) becomes a hopeless muddle story, unsatisfying ( |
| force himself on people and into situations that would make lesser men run for cover | would himself / people run for cover of situations and make force on lesser men |
| and unforgettable characters | unforgettable and characters |
| unfulfilling | unfulfilling |
| walked out muttering words like " horrible and " terrible , but had so much fun dissing the film that they did nt mind the ticket cost | walked out muttering words words like di fun the' ' mind the horrible filmbut had so much fun that they did tired, the terriblenssing ticket the film cost |

Table 6: Recovery examples for SST2 datasets with BERT$_{BASE}$ model.

