# OpenReview forum: "Beyond Gradient and Priors in Privacy Attacks: Leveraging Pooler Layer Inputs of Language Models in Federated Learning"
_ICLR.cc/2024/Conference — ICLR 2024 Conference Withdrawn Submission_

### Official Review · Reviewer_1dmy · 2023-10-26

**Soundness:** 1 poor
**Presentation:** 1 poor
**Contribution:** 1 poor
**Rating:** 1
**Confidence:** 3

**Summary:**

The authors propose a model poisoning data reconstruction attack on language models by first recovering intermediate activations.

**Strengths:**

- the topic of research of attacks on federated learning of language models is especially important given the recent surge of privacy concerns and large language models
- authors provide comparison with recent state of the art methods
- qualitative illustration of recoveries help understand better the relationship between ROUGE metrics and the reconstruction quality

**Weaknesses:**

Major.

The authors do not specify the threat model nor what their attack does in a high-level fashion, they only detail the specifics of the optimization tricks used but never the high-level flow: what do they observe ? what can they do/change ? etc. Considering the fact that authors seem to manipulate the architecture of the model, changing dimensions of layers and switching activation functions, the reviewer is guessing that the authors are doing model poisoning ("Malicious attacks"). But nowhere do the authors explain the general steps of the attack. Even when presenting related works, the positioning of this work with respect to the state of the art is not clear (especially wrt LAMP and Wang et al. 2023). Referencing Balunovic, Figure 1 or section 4 are not enough for readers to quickly understand the method's general principle and especially the setting in which the authors work.

The authors' contribution of "identify[ing] an issue with the gradient based attack: the gradient will be averaged in the context of large batch sizes and long sentences, thereby diluting ..." is not novel as do the many works on optimization-based data reconstruction attacks cited by the authors attest (DLG, GradInv, etc.). The contribution of targeting intermediate feature maps is not very novel either (see i.e. [1])

The authors' tone is too colloquial. A salient instance of this is the use of the "Revolutionary" adjective (or subtle/clever page 5) to qualify the authors' method. This qualifier is subjective. For instance the reviewer highly disagrees that the author's method is revolutionary. A scientific article should aim at remaining as neutral and objective as possible.

The authors describe their model poisoning attack as "subtle" (page 5), specifically they highlight a limitations of the work of Wang et al. 2023: the fact that Wang et al.'s attack is easy to detect. The reviewer does not understand how changing the model's architecture by adding orders of magnitude more dimensions and switching to non sparse activations is "subtle" or even more subtle than Wang's.

It is not clear to the reviewer why is the optimization objective of using the cosine similarity between recovered intermediate feature and Pooler input is even possible. Isn't the input of the Pooler layer exactly what we want to recover ?

The reviewer, in spite of being very familiar with the attack on gradients' literature and FL, has troubles understanding what the authors did therefore 1. the strong reject assessment 2. the short review and 3. the lack of comments on the results and their interpretation.


[1] Kariyappa, Sanjay, et al. "Cocktail party attack: Breaking aggregation-based privacy in federated learning using independent component analysis." International Conference on Machine Learning. PMLR, 2023.

**Questions:**

The reviewer encourages the authors to:
- add a paragraph on the threat model and a bird's eye view illustration of the method's in the setting described (more high-level than Figure 1)
- position their work precisely with respect to the related work specifically wrt LAMP and Wang et al. 2023: what are the innovations ?
- rework the text by 1. making it clearer what the method does and 2. removing most of the subjective statements related to the quality of the present work
- answer question on the optimization objective (see above)

---

> ### Author Response · Authors · 2023-11-17
>
> Thank you for your constructive feedback. Below are our responses to your main concerns. If these address your initial queries, we'd be grateful if you could revise your evaluation accordingly.
>
> ### Q1: Clarification of Threat Model
> ---
> We are thankful for the reviewer's astute observation that our discussion has been primarily focused on the optimization perspective without a high-level exploration of our threat model. This approach is indeed a result of deep contemplation.
>
> 1. **Classification of Attacks:**
> Following previous work, we categorized attacks as Malicious and Eavesdropping (or Honest-but-Curious Server) Attacks, but this oversimplifies the issue. `Traditional Malicious Attacks involve altering server states or model architecture, rendering gradient information useless. However, our method, which only modifies the Pooler layer of the model, doesn't affect the original training purpose, thus differing from typical Malicious Attacks.`
>
> 2. **Wider Pooler Layer:**
> The width of the Pooler Layer depends on the recovery needs. For a small input portion (X' with d_r << d_in), the required width reduces significantly, as detailed in Section 4: "`Flexibility of Recovered Dimension`". With the growth of Large Language Models like GPT-3, which already exceeds the necessary dimensions (12288 for d_r = 50), this width requirement becomes less vital. Moreover, our attack focuses on wide models, aligning with previous studies [Geiping et al.2020, Fowl et al.2022], revealing more risk in our settings.
>
> 3. **Activation Functions Change:**
> We've identified risks with activation functions like Selu, whose higher-order derivatives are neither odd nor even. Our ongoing research explores activation functions with strictly odd or even derivatives, like tanh or relu. While many previous studies have required specific activation functions [Zhu et al., 2019, Flow et al., 2022],  we view this kind of discovery as a contribution, not a drawback
>
> 4. **Redefining 'Honest-but-Curious':**
> Compared to Eavesdropping (Honest-but-Curious) attacks, our method does involve some modifications to the model. `However, we propose that 'honest' should not be narrowly defined as the server's absolute fidelity to clients. Instead, 'honest' should be understood as the server's adherence to its primary role of obtaining effective gradients and minimizing the training loss.` By this definition, our approach falls under this new category.
>
> 5. **Clarifying Our Method's Positioning:**
> In summary, our method cannot be classified as a Malicious or Eavesdropping Attack unless we redefine the Honest-but-Curious category. Our method, compared to Malicious Attacks, is more difficult to detect and does not violate the fundamental objective of federated learning. In contrast to Eavesdropping Attacks, our approach achieves more effective outcomes. Therefore, we mainly discuss our method from the perspective of optimization.
>
> 6. **Elaborating on the Threat Model:**
>     - We do not fine-tune the model's word and positional embeddings. The gradients for these embeddings are non-zero for words in the client's training data, enabling easy recovery of client sequences.
>     - The server operates under a redefined 'honest but curious' model, aiming to learn maximally from gradients without altering the learning protocol.
>
> 7. **Enhanced Understanding:**
> To aid the reviewer's understanding, we will revise our manuscript and include a table in the appendix. This table will delineate the differences between Malicious Attacks, Eavesdropping Attacks, and Honest-but-Curious Server attacks, with examples from previous work (LAMP, Wang et al., 2023).
>
> ### Q2: Novelty Concern
> ---
>
> Thank you for your comments on our work's novelty. We want to clarify a few points:
> 1. **Contribution Clarification:**
>     - Our contribution isn't about identifying gradient-based attack issues, `which was just an analysis foundation`. Our innovation is to `breach textual data privacy using the recovered information of intermediate features`.
>
> 2. **Differentiation from Prior Work [1]:**
>     - We recognize similar use of intermediate features in previous studies [1] on **image data**. However, our work pioneers this concept for **textual data attacks**, facing unique challenges not seen in image data due to differences in data representation and model architectures.
>     - Our technique method is totally different from [1]
>
> ### Q3: Tone and Subjectivity
> ---
> Thank you for pointing out the subjective tone. We will revise this. **The term "revolutionary" was specifically for our new metric on textual data information leakage, not our method (Contribution 4)**.
>
> ### Q4: Optimization Objective
> ---
> I appreciate your insight on our optimization objective. We aim to recover the `directional component` of the Pooler layer's input using decomposition, acknowledging challenges in reconstructing its magnitude. Our solution employs cosine similarity loss to address this limitation.
>
> Thank you again!

---

> ### Comment · Reviewer_1dmy · 2023-11-20
> **Acknowledgment**
>
> Q1: The reviewer disagrees with the author's characterization of "honest-but-curious" to qualify this work. Honest is a qualifier used primarily for data holding clients not for the aggregating sever.
> Q2: While applying a finding from a modality to another can qualify as a valid contribution the text should not be misleading.
> Q3: The reviewer disagree with the "revolutionary" qualifier even applied to the metric the authors found
>
> The reviewer stand with their original scoring.

---

> ### Author Response · Authors · 2023-11-20
>
> Dear Reviewer,
>
> Thank you for your initial review of our manuscript and your subsequent response.
>
> **We apologize for the delay in uploading our revised version, which might have led to it being overlooked**. Based on your valuable feedback, **we have significantly revised our manuscript**. We would greatly appreciate it if you could take the time to review our updated submission. We believe these changes comprehensively address your concerns and hope that you might reconsider your assessment in light of these updates.
>
> Again, Thank you for your valuable insights and the time you've dedicated to reviewing our work.
>
> Best regards,
> Authors

---

> ### Author Response · Authors · 2023-11-22
>
> Dear Reviewer,
>
>
> Thank you for your subsequent feedback. Below are our responses to your new concerns. If these address your initial queries, we'd be grateful if you could revise your evaluation accordingly.
>
>
> 1. **"Honest-but-Curious" Definition Misunderstanding:** We noticed a misinterpretation of the term 'honest' in the context of federated learning. In your review, 'honest' is associated with data-holding clients. However, in federated learning, as supported by a breadth of recent literature [1][2][3][4][5][6][7][8], 'honest' traditionally refers to the server. This distinction is crucial for understanding the methodology and assumptions of our study. Misinterpreting this term could lead to a skewed evaluation of our work, and we hope this clarification helps better understand our research approach.
>
>
> 2. **First Utilization of Intermediate Features for Privacy Text Attacks:** We wish to clarify our claim about pioneering the use of intermediate features in privacy attacks within NLP for federated learning again. The ICML 2023 paper [8] you referred to focuses on image data. **However**, our work is based on the domain of NLP and presents more complex challenges not encountered in CV, for example, the discrete nature of token ID and gradient average on the dimension of sequences. These challenges differentiate our approach from those applied to image data, underscoring the novelty and specificity of our work in the context of federated learning within NLP. **Moreover**, the methods used in these two works are different. (**We have consistently specified that our contributions are within NLP and federated learning, and at no point have we attempted to mislead regarding the scope of our work.**)
>
> 3. **Terminology Adjustment:** We would like to draw attention to a key revision in our manuscript that may have been overlooked. The term initially described as `Revolutionary Evaluation Metric` has been thoughtfully changed to `Semantic Evaluation Metric` to reflect its nature and scope more accurately.
>
> In summary, we hope these clarifications address the concerns raised and provide a clearer picture of the contributions and contexts of our work. We appreciate your attention to these matters and look forward to any further feedback you may have.
>
>
> Sincerely,
> Authors
>
> [1] Gupta, Samyak, et al. "Recovering private text in federated learning of language models." Advances in Neural Information Processing Systems 35 (2022): 8130-8143. (**NeurIPS 2022**)
>
> [2] Balunovic, Mislav, et al. "Lamp: Extracting text from gradients with language model priors." Advances in Neural Information Processing Systems 35 (2022): 7641-7654. (**NeurIPS 2022**)
>
> [3] Geiping, Jonas, et al. "Inverting gradients-how easy is it to break privacy in federated learning?." Advances in Neural Information Processing Systems 33 (2020): 16937-16947. (**NeurIPS 2020**)
>
> [4] Fowl, Liam, et al. "Decepticons: Corrupted transformers breach privacy in federated learning for language models." arXiv preprint arXiv:2201.12675 (2022). (**ICLR 2023**)
>
> [5] Boenisch, Franziska, et al. "When the curious abandon honesty: Federated learning is not private." 2023 IEEE 8th European Symposium on Security and Privacy (EuroS&P). IEEE, 2023.
>
> [6] Huang, Yangsibo, et al. "Evaluating gradient inversion attacks and defenses in federated learning." Advances in Neural Information Processing Systems 34 (2021): 7232-7241. (**NeurIPS 2021**)
>
> [7] Fowl, Liam, et al. "Robbing the fed: Directly obtaining private data in federated learning with modified models." arXiv preprint arXiv:2110.13057 (2021).(**ICLR 2022**)
>
> [8] Kariyappa, Sanjay, et al. "Cocktail party attack: Breaking aggregation-based privacy in federated learning using independent component analysis." International Conference on Machine Learning. PMLR, 2023.(**ICML 2023**)
>
> [9] Zhu, Ligeng, Zhijian Liu, and Song Han. "Deep leakage from gradients." Advances in neural information processing systems 32 (2019).  (**NeurIPS 2019**)

---

> > ### Comment · Reviewer_1dmy · 2023-11-22
> >
> > The reviewer disagrees strongly with the authors if clients are maliciously changing things such as activations number of neurones etc. it is NOT an honest-but-curious setup even though you could also look at the situation from the point of view of the server. Therefore it should be stated as such.

---

> > > ### Author Response · Authors · 2023-11-22
> > >
> > > Dear Reviewer,
> > >
> > > We greatly appreciate your prompt response and the insightful feedback provided. Your engagement is incredibly valuable and supportive to us. We would like to address your concerns regarding the "honest-but-curious" set up in our study:
> > >
> > > 1. **Clarification on the Adversary Role:** We understand there may be a misunderstanding regarding the adversary's role in our federated learning context. In our study, as reflected in all the cited references, the server is considered the attacker. This perspective is consistent across the literature on federated learning. Our discussion and analysis are thus focused on the server's viewpoint.
> > >
> > > 2. **Nature of the Attack in Federated Learning Protocol:** Even considering the server's perspective, our approach fundamentally differs from traditional malicious attacks. Under the federated learning protocol, our method does not violate key principles: **the generated gradients remain valid, and the model continues to train effectively with loss reduction**. This aspect sets our approach apart from conventional malicious attacks. In contrast to eavesdropping attacks, which only require parameters and gradients, our method involves modifications at a specific layer in deep neural networks, hence not fitting the typical eavesdropping category. We categorize our approach as 'honest-but-curious,' where 'honest' implies strict adherence to the federated learning protocol, which our method fully respects.
> > >
> > > 3. **Recognizing the Uniqueness of Privacy Attacks:** We also want to emphasize the special nature of privacy attacks. Any discovery that potentially reveals risks should not be overlooked. Attackers often strive to meet attack conditions without concern for the category of attack used. Our method, when appropriately designed, is less detectable compared to malicious attacks and more effective than eavesdropping attacks. Its value in unveiling potential risks in federated learning environments should not be underestimated.
> > >
> > > These clarifications will assist in better understanding the unique aspects and contributions of our work. Thank you for considering our perspective.
> > >
> > > Sincerely,
> > > Authors

---

### Official Review · Reviewer_eYy2 · 2023-10-29

**Soundness:** 3 good
**Presentation:** 2 fair
**Contribution:** 3 good
**Rating:** 6
**Confidence:** 4

**Summary:**

Privacy attacks could extract sensitive information from large language models (LLM) in federated learning (FL), either with limited batch size or being detectable. The authors aim to recover texts in various batch-size settings yet challenging to detect with the constraint of LLM  equipped with a unique Pooler layer.

The solution is to recover the intermediate feature to provide enhanced supervisory information.  This Pooler layer captures a comprehensive representation of the input text. A two-layer neural network-based reconstruction technique is used to retrieve the inputs destined for this layer meticulously. The method provides a continuous supervisory signal, offering additional feature-level guidance that assists optimization-based attacks.

By combining gradient inversion and prior knowledge, the proposed approach achieves better results on different datasets, tailored with different batch sizes (e.g., 1,2,4,8).

**Strengths:**

+ The idea works well in success rate and improves existing attacks. Also, it is absorbing from my perspective.
+ Extract a unique and concise problem to solve in LLM privacy.
+ The first to suggest utilizing intermediate features as continuous supervised signals.

**Weaknesses:**

- The presentation should be improved. [see Question 3, Question 4, Question 5, Question 6]
- Experimental settings. [see Question 7]
- Require more clear discussion/comparison on related works. [see Question 1, Question 2]

**Questions:**

1. Could the authors illustrate more related works on intermediate features? For example, [HKHG+] proposes to ''examine the representations in intermediate feature maps ... ''.
The authors claim, "The first to suggest utilizing intermediate features...".  I feel a little confused about the "the first" here. Could the authors elaborate on it?

2. In Section 2.2, "Nonetheless, numerous studies have highlighted the risks associated with textual information." Could the authors explain more about the conclusions/findings that have been studied? What are explicit risks specific to LLM textual information? Given my understanding of the abstract, the authors target solving hurdles of attacks ("extracting sensitive data from LLM in federated learning"). Is any additional challenge introduced in the *federated LLM* compared with LLM/FL?

3. What is "[CLS]" in the introduction?

4. Could the authors detail the security model? In federated learning, clients and a server exist in common. For the proposed attack, who is the adversary, and what is the adversary's ability? What is the assumption of all participants in FL? If the adversary uses the intermediate features, does it mean the adversary has more knowledge (i.e., weaker security assumption) than previous attacks? What is the explicate attacking goal?

5. Many notations in Section 3 are missing. For example, what is the hat in the equation 3? In Equation 2, $B$ is suddenly used. What are "(3)" and $\otimes 3$ in Equation 5?

6. Section 4 provides many optimization signals. Could the authors bridge the experimental findings and theoretical conclusions in (Wang et al., 2023)? Could the authors give a high-level walkthrough of various optimizations?

7. How do authors compare with previous arts in the experiments? For example, Table 1 shows better results with different batch sizes. In the abstract, the authors point out that previous works have limited batch size. Continuously, could the authors explain more about why the previous works become worse when enlarging the batch size?


[HKHG+] Enhancing Adversarial Example Transferability with an Intermediate Level Attack. Qian Huang, Isay Katsman, Horace He, Zeqi Gu, Serge Belongie, and Ser-Nam Lim (ICCV' 2019)

**Details Of Ethics Concerns:**

N.A.

---

> ### Author Response · Authors · 2023-11-18
>
> Thank you for your constructive feedback. Below are our responses to your main concerns. If these address your initial queries, we'd be grateful if you could revise your evaluation accordingly.
>
> ### Q1: Clarification on the Novelty of Using Intermediate Features
> ---
>
> We sincerely appreciate your reference to the paper [HKHG+], which shares a similar intuition of shifting the focus from model input to intermediate features but in a completely different context.
>
> 1. **Different Context and Application:** Our work uniquely focuses on recovering textual data in LLMs within the context of federated learning (FL). The cited work [HKHG+] primarily examines the use of intermediate feature perturbation to enhance the transferability of adversarial inputs in the context of black-box adversarial attacks.
>
> 2. **Totally Different Methods**: Our method does not know the true intermediate feature, instead utilizing gradient information and tensor decomposition technique to estimate the feature. The reference [HKHG+] can easily calculate the intermediate features with the model and input.
>
> To better respond to the reviewer's concern, we will present a discussion that delves into the underlying intuitions of both approaches, highlighting the differences and acknowledging the similarities where applicable.
>
> ### Q2: Risks in LLMs within FL
> ---
>
> Thank you for inquiring about the privacy risks of textual data in LLMs within FL. To safeguard user privacy, users' textual data is typically stored on local edge devices. These devices compute gradient information and upload it to the central server. Ideally, the server doesn't access the original text, but studies have shown it's possible to reconstruct training text from gradient information and prior knowledge. We hope this clarifies the challenges in LLMs within FL.
>
> ### Q3 & Q5: Clarification on Notations
> ---
>
> 1. **[CLS] Token:** This special token, added at the start of each input sentence, is selected to represent the sentence for classification tasks.
>
> 2. **$\hat{x}$ and $\hat{y}$ in Equation 3:** These symbols are the optimization goal that represents the estimated input and label, respectively.
>
> 3. **B in Equation 2:** Refers to Batch Size.
>
> 4. **$\nabla^{3}g(w)$ and $x_{i}^{\otimes3}$:** $\nabla^{3}g(w)$ is the third derivative of $g(w)$, and $x_{i}^{\otimes3}$ signifies the tensor product of vector $x_{i}$ with itself three times.
>
> ### Q4: Clarification on security model
> ---
>
> 1. **Adversary in the Attack:** The server in FL is the adversary, operating under a redefined 'honest but curious' setting. It aims to learn as much text as possible from the gradients without altering the learning protocol. (`The honest here refers to the server always minimizing the training loss without other objectives`)
>
> 2. **Adversary's Ability:** The server can modify the Pooler layer of the model, allowing it first to recover the intermediate features (`No other prior knowledge`). However, the server does not finetune the word and positional embeddings (The gradients for these embeddings are non-zero for words in the client's training data, enabling easy recovery of client sequences).
>
> 3. **Action of Participants:** Clients compute the gradient and upload it to the server, which is benign.
>
> 4. **Security Implication:** This approach challenges traditional security models in FL. It's less detectable than Malicious Attacks and more invasive than typical Honest-but-Curious attacks, aiming to recover textual data from shared gradients. (`The honest here refers to the server not changing model architecture and parameters`)
>
> 5. **Attacking Goal:** The primary goal is to recover client sequences or information from the gradients `without deviating from the fundamental objectives of federated learning, like effective gradient aggregation and model loss minimization`.
>
> ### Q6: Theoretical Discussion.
> ---
>
> We have addressed the reviewer's concern in the following sections:
>
> 1. **Bridging Experimental Findings with Theory:** In Section 5.3, "`Impact of Recovery Dimension`" and Appendix Section A.4, "`Influence of Data Dependence`" we have thoroughly discussed how our experimental findings correlate with the theoretical insights presented in (Wang et al., 2023).
>
> 2. **Discussion on Optimization Signals:** We have also explored the impact of different optimization signals in `the second and third paragraphs` of Section 5.3. Additionally, our ablation experiments in Section 5.4, “`Impact of Feature Match in Different Optimization Phases`”, further elaborate on how varying optimization signals influence attack performance.
>
> ### Q7：Comparison with Previous
> ---
>
> Please refer to Section 5.3, "`Results and Analysis`", for a detailed explanation. The performance degradation observed in previous works when increasing the batch size is attributed to the gradient being averaged over more tokens and sentences. Our method introduces an additional supervisory signal that mitigates this issue.
>
> Thank you again!

---

> > ### Comment · Reviewer_eYy2 · 2023-11-21
> >
> > Thanks for the response. I would like to keep my rating.

---

### Official Review · Reviewer_GEoJ · 2023-10-31

**Soundness:** 3 good
**Presentation:** 3 good
**Contribution:** 3 good
**Rating:** 6
**Confidence:** 3

**Summary:**

This paper focuses on data extraction attack against large language model in the federated learning setting. The authors propose a novel attack by leveraging the input of the Pooler layer of language models to offer additional feature-level guidance that effectively assists optimization-based attacks. Evaluations on benchmark text classifcation datasets demonstrate the effectiveness of the proposed method with different batch sizes and models.

**Strengths:**

- important research topic
- novel attack methodology
- well-structured paper

**Weaknesses:**

- only evaluate on text classification
- lacking cost analysis
- possible countermeasures are needed

**Questions:**

- The authors demonstrate the effectiveness of the proposed method on text classification tasks. However, it is unclear how well it would perform on other types of tasks.

- I appreciate the authors' effort on presenting the superior performance of the attack. In addition, a cost analysis (time and resource) would be good to understand the trade-off on different attacks.

- This paper does a great job in presenting a powerful attack. The authors are suggested to discuss (and better evaluate) possible countermeasures.

---

> ### Author Response · Authors · 2023-11-19
>
> Thank you for your constructive feedback. Below are our responses to your main concerns. If these address your initial queries, we'd be grateful if you could revise your evaluation accordingly.
>
> ### Q1：Performance on Other Types of Tasks
> ---
>
> Thank you for highlighting the importance of evaluating our proposed method's effectiveness in other natural language processing tasks, such as translation or generation tasks. We acknowledge that our approach might face greater challenges in these contexts. In such tasks, the loss generated by different tokens is typically summed or averaged before computing gradients, in contrast to classification tasks that usually produce a single loss.
>
> 1. **Acknowledging Shared Challenges in the Community:**： We recognize that these challenges are not unique to our method but are common obstacles faced across the community in privacy attack research. This issue underscores the complexity and variety of tasks in natural language processing and their implications for privacy attacks.
>
> 2. **Emphasizing the Practical Importance of Privacy Attack Research:**： We wish to emphasize that our primary goal is not to develop a universal privacy attack method but to highlight potential threats of LLMs within federated learning. It's essential to understand these vulnerabilities of the current protocol to prevent potential risks.
>
> 3. **Future Research Directions:**: Your concern aligns with our future research direction. We are motivated to explore how our attack methodology can be adapted or improved to address the unique challenges in tasks like translation and generation.
>
> ### Q2: Cost Analysis
> ---
>
> We appreciate your attention to the cost implications of our method and would like to clarify two key aspects regarding this:
>
> 1. **Comparable Cost to Traditional Optimization-Based Techniques:**: Our method incurs almost the same computational cost as other traditional optimization-based techniques. This is primarily because the recovery of intermediate features in our approach involves a tensor decomposition technique, which is a one-time process. The computational effort for this part is negligible compared to the subsequent optimization steps. Moreover, during the optimization phase, our method only requires the additional computation of a cosine similarity loss, which also does not significantly increase the computational burden. Based on our assessments, our method does not add substantial extra computational overhead (less than 1.1 times the usual cost).
>
> 2. **Context of Attack Scenarios in Practice:**: It's important to consider the practical context of attack scenarios. Attackers often prioritize the effectiveness of the attack over the cost, implying that computational expense is not the primary concern in such research. This perspective is crucial in understanding the feasibility of privacy attacks, including ours, in real-world scenarios.
>
> ### Q3: Possible Countermeasures
> ---
>
> We are grateful for your attention to both the attack methodologies and potential defense techniques in our study. We recognize the importance of defenses in the context of privacy attacks and appreciate this opportunity to discuss them.
>
> 1. **Effectiveness of Attack in Certain Settings:** Our research acknowledges that honest-but-curious type attacks on LLMs, especially when freezing the token and positional embeddings, tend to be effective primarily in settings with smaller batch sizes. While our method significantly improves the effectiveness of attacks for B > 1, we observe a notable decline in effectiveness when B > 8. This suggests that in federated learning scenarios with large batch sizes and a benign server, textual data privacy can still be reasonably safeguarded. (`The 'honest' here refers to the server's goal to recover client sequences or information from the gradients without deviating from the fundamental objectives of federated learning, like effective gradient aggregation and model loss minimization.`)
>
> 2. **Decision Against Including Defense Techniques:** Based on this observation, we chose not to focus on defense techniques within the scope of this paper. Our primary objective was to reveal the potential risk of privacy breaches in LLMs within federated learning settings, particularly those with moderate batch sizes.
>
> Thank you again!

---

> > ### Comment · Reviewer_GEoJ · 2023-11-22
> >
> > Thanks for the clarification. I would keep the score as it is.

---

### Official Review · Reviewer_7zQ4 · 2023-11-01

**Soundness:** 3 good
**Presentation:** 3 good
**Contribution:** 2 fair
**Rating:** 6
**Confidence:** 3

**Summary:**

This paper proposes a new attack method to enhance the text recovery rate of language models under the Federated Learning setting. It is based on two techniques: (1) leveraging gradient data and prior knowledge to extract sensitive information (Zhu et al., 2019; Deng et al., 2021; Balunovic et al., 2022; Gupta et al., 2022), and (2) Two-Layer Neural Network-Based Reconstruction (Wang et al., 2023), whose results will be used as the prior knowledge. By combining these two techniques together, the proposed method tries to address existing challenges in enhancing the recovery rate of text in larger batch-size settings while being hard to detect and defend against. This paper compares the proposed method with existing baseline methods and proves its superiority.

**Strengths:**

1. This paper studies a very interesting problem, which is the attacks on language models under the Federated Learning setting.
2. The proposed methods achieve better results than existing baselines.

**Weaknesses:**

1. The proposed method is based on the existing method and applies it to the language model under the federated learning setting, which fits the setting of the existing method well. The contribution is limited.
2. This paper does not solve the batch size issue efficiently. Since the batch size (i.e., 8) that the proposed method can work well on is still very small compared to common settings for batch sizes.
3. This paper does not demonstrate how existing defense methods work to defend against the proposed attack, or in other words, how the proposed attack performs against the defense methods.

**Questions:**

1. Does this method require the attacker to know the attacked model's structure, such as whether it has a Pooler layer or not, as a priori?
2. In the text domain, what kind of information is considered private? For instance, if there's a phrase 'this food is … ,' and then 'delicious' is recovered, is this considered private information?

---

> ### Author Response · Authors · 2023-11-19
>
> Thank you for your constructive feedback. Below are our responses to your main concerns. If these address your initial queries, we'd be grateful if you could revise your evaluation accordingly.
>
> ### Q1: Contribution and Novelty of the Proposed Method
> ---
>
> We appreciate your acknowledgment of our work's structure and ideas. However, it's crucial to emphasize that our approach is not a mere combination of existing methods; it includes several unique contributions:
>
> 1. **Theoretical vs. Practical Application:** Wang et al.'s work, while theoretically effective, encounters practical challenges due to various constraints that limit its effectiveness in real-world scenarios. In contrast, our approach adapts these theoretical insights to practical federated learning settings.
>
> 2. **Limitations in Wang et al.'s Work:** Wang et al.'s method is most effective in a two-layer neural network context. Extending it beyond this depth requires customizing model parameters, like identity matrices, which contradicts the definition of 'honest but curious' in federated learning. Our method maintains this 'honesty' by focusing on the top two layers of deep neural networks. (`The 'honest' here refers to the server's goal to recover client sequences or information from the gradients without deviating from the fundamental objectives of federated learning, like effective gradient aggregation and model loss minimization.`)
>
> 3. **Focus on Intermediate Feature Recovery:** Unlike Wang et al.'s work, which is limited to directly recovering model input, our method explores the potential of recovering intermediate features, adding a new dimension to attack strategies.
>
> 4. **Flexibility in Recovery Dimension:** Our approach offers more flexibility in optimizing recovery dimensions, as discussed in Section 4.2 "Flexibility of the Recovery Dimension." We can adaptively try different dimensions to meet specific needs, which is not the case in Wang et al.'s work.
>
> 5. **Strategic Weight Initialization:** We address the issue of gradients approaching zero due to weight randomization, as often happens in Wang et al.'s method. In Section 4.2 "Strategic Weight Initialization," we innovatively propose a coexistence of original and random weights, effectively overcoming this challenge.
>
> 6. **Addressing Data Point Order in Batches:** Wang et al.'s method struggles with determining the order of data points in the same batch. Our method, from an attacker's perspective, compares with the true features and adopts a greedy strategy based on the highest cosine similarity for one-to-one correspondence, effectively resolving the data point order issue.
>
>
> ### Q2 & Q3: Batch Size and Defense Methods
> ---
>
> We recognize the importance of defenses in the context of privacy attacks and appreciate this opportunity to discuss them.
>
> 1. **Effectiveness of Attack in Certain Settings:** Our research acknowledges that honest-but-curious type attacks on LLMs, especially when freezing the token and positional embeddings, tend to be effective primarily in settings with smaller batch sizes. While our method significantly improves the effectiveness of attacks for B > 1, we observe a notable decline in effectiveness when B > 8. This suggests that in federated learning scenarios with large batch sizes and a benign server, textual data privacy can still be reasonably safeguarded.
>
> 2. **Decision Against Including Defense Techniques:** Based on this observation, we chose not to focus on defense techniques within the scope of this paper. Our primary objective was to reveal the potential risk of privacy breaches in LLMs within federated learning settings, particularly those with moderate batch sizes.
>
> ### Q4: Knowledge of Model Architecture
> ---
>
> Thank you for raising the question about the attacker's knowledge of the model architecture; we clarify this as follows:
>
> 1. **Knowledge of Model Architecture in Security Settings:**
>    - In the security model we've considered, whether the server setting is honest or malicious, the model's architecture is always known to the server. This knowledge is an inherent aspect of the federated learning setup, where the server orchestrates the learning process and, therefore, has access to the model architecture.
>
> 2. **Implications for Potential Hackers:**
>    - Furthermore, even in scenarios where a hacker might steal model parameters, the limited variety of commonly used language models today often makes it possible to infer or guess the model architecture.
>
> ### 5：Definition of Private Information in Text
> ---
>
> Thank you for your question. In your example, while 'delicious' isn't inherently sensitive, the key point is that attack methods don't distinguish between words like 'delicious' and genuinely sensitive information like personal health data. If our attack can recover something as benign as 'delicious,' it also has the potential to recover more sensitive information. Therefore, any potential data recovery is considered a privacy risk

---

### Official Review · Reviewer_tBfV · 2023-11-01

**Soundness:** 3 good
**Presentation:** 3 good
**Contribution:** 2 fair
**Rating:** 3
**Confidence:** 3

**Summary:**

This paper studies the gradient inversion attack, that is to reconstruct the data from the model gradient. The paper focuses on the attack for language models such as BERT. The key idea is to recover the intermediate feature, the input before the pool layer. Then, it applies a previous gradient-matching attack but adds feature matching to the loss. It empirically shows that with this additional feature matching loss, the recovery accuracy can be improved.

**Strengths:**

1. The presentation and clarity are generally great.
2. The studied problem is to validate an important privacy vulnerability, which motivates the study of privacy.
3. The evaluation is systematic. The proposed methods together with baselines are evaluated cross 3 benchmark datasets and various batch sizes.

**Weaknesses:**

My main concern is that the proposed attack only works with some constraints: certain types of activation, a large enough hidden dimension of the Pooler layer, and random initialization for most variables in the Pooler layer. These constraints seem to deviate from the popular design in the usage of large language models.

In Table 1, "Ours" is evaluated with two different activations in the architecture, SELU and $x^3+x^2$. Then, it is not clear to me what architecture was used for baseline evaluation. Should each baseline also have two rows corresponding to two activations?

It is not well explained why each unique $x_i$ can be reconstructed by Equation 5&6 illustrated on page 4. Especially, when batch size $B$ is large enough, if I understand it correctly, it is possible that there are multiple solutions for $x_i$.

**Questions:**

Please see the "Weaknesses".

---

> ### Author Response · Authors · 2023-11-18
>
> Thank you for your constructive feedback. Below are our responses to your main concerns. If these address your initial queries, we'd be grateful if you could revise your evaluation accordingly.
>
> ### Q1. Constraints of the Proposed Attack:
> ---
>
> We recognize some limitations to a certain extent, but not entirely, as real-world attacks are often more extreme than this scenario. To provide further clarity, we compare our methodology from an attack perspective with prior research.
>
> 1. **Wide Pooler Layer Constraints:** The width of the Pooler Layer is adaptive to recovery needs. As we detailed in Section 4, "`Flexibility of Recovered Dimension`", for smaller input portions (X' with d_r << d_in), the required width is significantly reduced. With advancements in Large Language Models like GPT-3, which already surpass necessary dimensions (12288 for d_r = 50), this width requirement becomes less critical. Furthermore, our focus on wide models aligns with previous studies (e.g., Geiping et al., 2020; Fowl et al., 2022), highlighting increased risks in our settings.
>
> 2. **Activation Functions Change:** We have identified risks related to using activation functions such as Selu, whose higher-order derivatives are neither odd nor even. Our ongoing research extends to activation functions with strictly odd or even derivatives, such as Tanh or Relu. While many previous studies have focused on specific activation functions (Zhu et al., 2019, Flow et al.2022), we view this discovery as a contribution, not a drawback.
>
> 3. **`Comparison with Previous Studies`:**
>     - Previous works [e.g., Flow et al., 2022, 2023; Boenisch et al., 2023] have implemented attacks capable of handling very large batch sizes and long sequence text recovery. However, these methods, called Malicious Attacks, have critical flaws. Firstly, they violate the federated learning protocol, as the gradients returned by the client during an attack are meaningless. Secondly, they involve multi-layer modifications to model parameters or even insert new modules, which are far more aggressive than our modifications. In contrast, we only modify the Pooler Layer and consistently adhere to the federated learning protocol. Nevertheless, we recognize the value of these works in advancing privacy attack research.
>     - Works like LAMP can extract private text information using only optimization-based techniques and prior knowledge without any model modifications. However, this is a self-imposed limitation. In reality, servers can undertake actions well beyond this scope. Although such actions may be more easily detectable, our method achieves less detectability while providing additional optimization signals, marking a significant innovation.
>
> 4. **`Specificity of Attack Scenarios`:** Research in privacy attacks holds a unique importance in practice. This is because every discovery in this area, even with certain constraints, can cause tremendous destruction once happens. The attacker will try their best to satisfy the constraints because of the hidden massive profit. Therefore, the risks uncovered by these studies are often difficult to measure, making them crucial for understanding security vulnerabilities.
>
> ### Q2. Architecture of Evaluation Baselines
> ---
>
> We are grateful for your observation regarding the architecture used in our baseline evaluations, and we would like to clarify this aspect to dispel any confusion.
>
> In our comparative analysis, we maintained the original model structures of our baselines without altering their activation functions. This decision was driven by our goal to uncover potential private text attack methods, where the choice of activation function plays a crucial role. `In our research, we discovered that activation functions with neither purely odd nor even higher-order derivatives, such as Selu or $x^3+x^2$, pose significant risks in our setting compared to other functions like Relu/Tanh`.
>
> We recognize the discovery regarding the choice of activation functions **is a part of our contribution**. It highlights the vulnerabilities in real-world applications in this setting. Therefore, we do not consider the selection of activation functions as a distilled item for comparison with other methods. Instead, it is more pertinent to internal comparisons within our own methodology. We have included a detailed discussion on this topic in section A.3 "`Impact of activation function`".
>
> ### Q3. Clarification on Equations 5 & 6
> ---
>
> We appreciate your request for clarification regarding Equations 5 & 6, and we will provide more explanation in the appendix of our revised manuscript.
>
> Regarding the uniqueness of $x_i$'s recovery, it largely depends on the correlation among different data points within the batch. In the appendix, we have a dedicated discussion section titled "`Influence of Data Dependence`", where we delve into how the cosine similarity between recovered features and actual features gradually decreases as batch size increases.

---

> > ### Comment · Reviewer_tBfV · 2023-11-23
> > **Official Comment by Reviewer tBfV**
> >
> > Thank the authors for their clarification.
> >
> > 1. Although both "Wide Pooler Layer Constraints" and "Activation Functions Change" have been studied in the literature separately, the combination of them further limits the realistic applications.
> > 2. The modification of the architecture is not the power of adversary, but the setting of the FL. Therefore it is not fair to compare the proposed attack with the baselines for different architectures.

---

> ### Author Response · Authors · 2023-11-23
>
> Dear Reviewer:
>
> We extend our sincere thanks for your latest response. Your insights, particularly from the standpoint of **research that makes life better**, are well-token.
>
> However, we kindly urge the reviewer to consider our work from the perspective of **research that makes life worse** for the last time. Our study specifically focuses on identifying and understanding critical risks associated with Large Language Models (LLMs) in the context of federated learning. There may exist certain limitations in our methodology, but the significant risks and dangers that our research uncovers are both actual and consequential.
>
> Thank you for your reinvested time and comments
>
> Best,
> Authors.

---

### Author Response · Authors · 2023-11-20
**Revision Summary & Common Response to all Reviewers**

Dear Reviewers,

We sincerely appreciate your insightful feedback and have thoroughly revised our manuscript to address your concerns (modifications are highlighted in red and blue). If these adjustments adequately address your initial queries, we would be grateful if you could kindly update your evaluation accordingly.

Below is a summary of the critical modifications we have implemented:

1. **Clarification on Threat and Security Models:** Responding to Reviewers `1dmy` and `eYy2`, we've clarified our threat and security models, as well as the high-level categorization of our method. These clarifications are now included in the introduction (`highlighted in red and blue in the second paragraph`) and in the related work section (`in the last two paragraphs, also highlighted`). Furthermore, a systematic discussion of these models is provided in `Appendix-A`.

2. **Distinction from Previous Studies:** To address concerns from Reviewers `1dmy` and `tBfV` about how our work differs from previous studies, we've made relevant modifications `in the sections mentioned above`. Additionally, we have included a comparative analysis in `Appendix-A`, presented in a tabular format.

3. **Discussion on Method Constraints:** For Reviewer `tBfV`'s query about method constraints, we have added detailed discussions and responses in `Related work`, `Appendix A-1`, and `Apendix A-2`.

4. **Explanation of Two-Layer Neural Network-Based Reconstruction Method:** Addressing queries from Reviewers `1dmy` and `tBfV` regarding the two-layer-neural-network-based reconstruction method, we have provided an in-depth explanation and justification in `Appendix C-1`.

5. **First Use of Intermediate Feature in Privacy Textual Data Attack:**  In response to Reviewers `eYy2` and `1dmy`, we have elaborated in `Appendix C-2` on whether our introduction of the intermediate feature constitutes a first in privacy textual data attack.

6. **Objectivity in Presentation:** Reviewer `1dmy`'s concern about objectivity has been addressed by highlighting changes in the manuscript, `focusing on the sections emphasized by the reviewer`.

7. **Clarification of Notations:** To clarify the notation issues raised by Reviewer `eYy2`, we have made modifications in the relevant sections, indicated by highlighted text.

8. **Rationale Behind Using Cosine Similarity in Optimization:** Lastly, in response to Reviewer `1dmy`'s question about using cosine similarity in optimization, we have provided a comprehensive explanation at the end of `Appendix C-1`.

9. **Direct Responses on OpenReview:** For some inquiries that were not explicitly reflected in the revised version of our manuscript, we invite reviewers and readers to view our `direct responses on OpenReview`. We have provided detailed explanations and justifications there to address these specific queries, ensuring comprehensive coverage of all concerns raised.

We believe these revisions significantly enhance the clarity, rigor, and contribution of our work. We thank you for your constructive feedback, which has been invaluable in improving the quality and impact of our research.

---

### Author Response · Authors · 2023-11-22
**Gentle Reminder: Feedback on Our Revision and Rebuttal**

Dear Reviewers,

I hope this message finds you well. We understand the demands of your role and genuinely appreciate the time and expertise you've already invested in reviewing our paper.

**As our discussion deadline is approaching in 1 day, we wanted to remind you of our revision and rebuttal gently**. Your feedback on our responses is vital to us, and we would sincerely appreciate any additional insights or clarifications you might have.

If you find our rebuttal satisfactory and believe the paper has been improved based on your invaluable feedback, we kindly request you consider revising the evaluation. Such an adjustment would be deeply appreciated, underscoring the importance of your feedback on our work.

Warm regards,

Authors

---

### Author Response · Authors · 2023-11-22
**Gentle Reminder: Feedback on Our Revision and Rebuttal**

Dear Reviewers:

I hope this message finds you well. We understand the demands of your role and genuinely appreciate the time and expertise you've already invested in reviewing our paper.

**As the discussion phase is approaching its closure, with just 12 hours remaining, we are eager to hear your thoughts and comments, especially from reviewers haven't response to our rebuttal and new manuscript**

If you find our rebuttal or revision satisfactory and believe the paper has been improved based on your invaluable feedback, we kindly request you consider revising the evaluation. Such an adjustment would be deeply appreciated, underscoring the importance of your feedback on our work.

Warm regards,
Authors